# CONTROLLING VISION-LANGUAGE MODELS FOR MULTI-TASK IMAGE RESTORATION

**Ziwei Luo, Fredrik K. Gustafsson, Zheng Zhao, Jens Sjölund, Thomas B. Schön**
Department of Information Technology, Uppsala University
{ziwei.luo,fredrik.gustafsson,zheng.zhao}@it.uu.se
{jens.sjolund,thomas.schon}@it.uu.se

## ABSTRACT

Vision-language models such as CLIP have shown great impact on diverse downstream tasks for zero-shot or label-free predictions. However, when it comes to low-level vision such as image restoration their performance deteriorates dramatically due to corrupted inputs. In this paper, we present a degradation-aware vision-language model (DA-CLIP) to better transfer pretrained vision-language models to low-level vision tasks as a multi-task framework for image restoration. More specifically, DA-CLIP trains an additional controller that adapts the fixed CLIP image encoder to predict high-quality feature embeddings. By integrating the embedding into an image restoration network via cross-attention, we are able to pilot the model to learn a high-fidelity image reconstruction. The controller itself will also output a degradation feature that matches the real corruptions of the input, yielding a natural classifier for different degradation types. In addition, we construct a mixed degradation dataset with synthetic captions for DA-CLIP training. Our approach advances state-of-the-art performance on both *degradation-specific* and *unified* image restoration tasks, showing a promising direction of prompting image restoration with large-scale pretrained vision-language models. Our code is available at https://github.com/Algolzw/daclip-uir.

## 1 INTRODUCTION

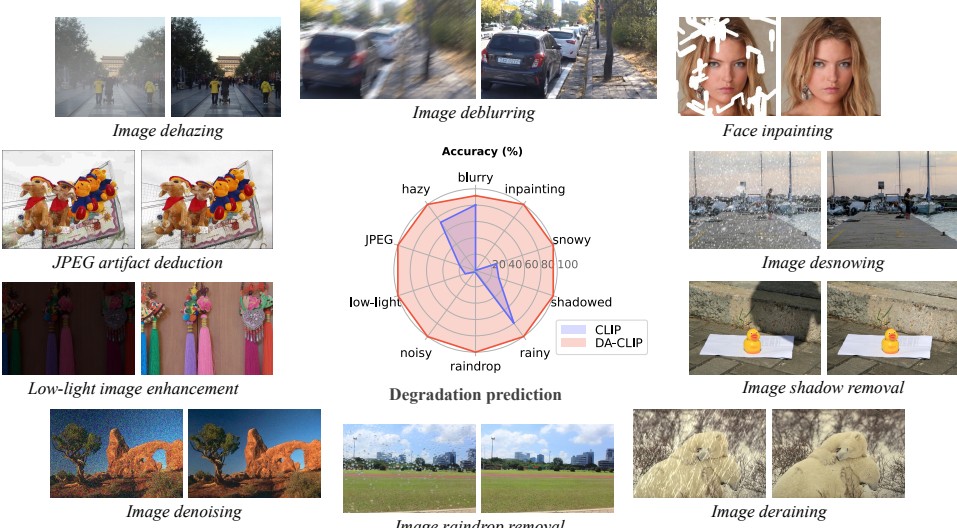

Figure 1: This paper leverages large-scale pretrained vision-language models for multi-task image restoration. Compared to CLIP (Radford et al., 2021), our approach precisely predicts the degradation embeddings for corrupted inputs and also outputs high-quality features for better image restoration performance. For all examples above, the right results are produced by our single *unified* model.

Large-scale pretrained vision-language models (VLMs) such as CLIP (Radford et al., 2021) have recently garnered significant attention, in part because of their wide-reaching usefulness on many fundamental computer vision tasks (Gu et al., 2021; Zhang et al., 2022; 2023). However, existing VLMs have so far had limited impact on low-level vision tasks such as image restoration (IR), presumably because they do not capture the fine-grained difference between image degradation types such as "blurry" and "noisy" (Ni et al., 2023). Consequently, the existing VLMs often misalign image features to degradation texts. This is not surprising, considering that VLMs are generally trained on diverse, web-scale datasets, in contrast to most image restoration models which are trained on comparatively small datasets that are curated for a specific task without corresponding image-text pairs (Li et al., 2019; Zhang et al., 2017; Zamir et al., 2022).

Image restoration methods often simply learn to generate images pixel-by-pixel without leveraging task knowledge, which usually requires repeated training of the same model for specific degradation types. A recent line of work has, however, focused on *unified* image restoration, where a single model is trained on a mixed degradation dataset and implicitly classify the type of degradation in the restoration process (Li et al., 2022a; Potlapalli et al., 2023). While the results are impressive, they are still limited to a small number of degradation types and the specific datasets that go with them. In particular, they do not make use of the vast amount of information embedded in VLMs.

In this paper, we combine the large-scale pretrained vision-language model CLIP with image restoration networks and present a multi-task framework that can be applied to both *degradation-specific* and *unified* image restoration problems. Specifically, aiming at addressing feature mismatching between corrupted inputs and clean captions, we propose an *Image Controller* that adapts the VLM's image encoder to output high-quality (HQ) content embeddings aligned with clean captions. Meanwhile, the controller itself also predicts a degradation embedding to match the real degradation types. This novel framework, which we call degradation-aware CLIP (DA-CLIP), incorporates the human-level knowledge from VLMs into general networks that improve image restoration performance and enable unified image restoration. In addition, to learn high-quality features and degradation types from low-quality (LQ) inputs, we construct a large mixed-degradation dataset for ten different image restoration tasks based on BLIP (Li et al., 2022b). As shown in Figure 1, our DA-CLIP accurately classifies the ten different degradation types and can readily be integrated into existing restoration models, helping produce visually appealing results across the different degradations.

Our main contributions are summarised as follows: **(1)** We present DA-CLIP to leverage large-scale pretrained vision-language models as a universal framework for image restoration. The key component is an image controller that predicts the degradation and adapts the fixed CLIP image encoder to output high-quality content embeddings from corrupted inputs. **(2)** We use cross-attention to integrate the content embedding into restoration networks to improve their performance. Moreover, we introduce a prompt learning module to better utilize the degradation context for unified image restoration. **(3)** We construct a mixed degradation dataset containing ten different degradation types with high-quality synthetic captions. This dataset can be used to train either DA-CLIP or a unified image restoration model. **(4)** We demonstrate the effectiveness of DA-CLIP by applying it to image restoration models for both degradation-specific and unified image restoration. Our approach achieves highly competitive performance across all ten degradation types.

## 2 BACKGROUND AND RELATED WORK

**Image restoration** Image restoration aims to recover a high-quality image from its corrupted counterpart, which is a fundamental and long-standing problem in computer vision and contains various tasks such as image denoising (Zhang et al., 2017), deraining (Ren et al., 2019), dehazing (Song et al., 2023), deblurring (Kupyn et al., 2018), etc. Most existing works focus on the *degradation-specific* task which trains multiple models for different tasks separately by directly learning the target with common pixel-based losses (e.g., $\ell_1$ or $\ell_2$). Recently, however, increasing attention has been paid to *unified* image restoration where multiple image restoration tasks are handled with a single model (Li et al., 2022a; Potlapalli et al., 2023). These works achieve impressive results by using an additional encoder (Li et al., 2022a; Zhou et al., 2022) or a visual prompt module (Potlapalli et al., 2023) to implicitly cluster inputs according to the degradation type. However, they are still limited to a few image restoration tasks and do not consider auxiliary information about the degradation type.

Figure 2: Overview of our method. DA-CLIP freezes both the text and image encoders of a pre-trained CLIP but learns an additional image controller with contrastive learning. This controller predicts degradation features to match real corruptions and then controls the image encoder to output high-quality content features. Once trained, DA-CLIP can be integrated into other image restoration models by simply adding a cross-attention module and a degradation feature prompting module.

**Blind image restoration (BIR)** To deal with unknown degradation levels, BIR comes into view and has shown promising results for photos captured in the real-world (Zhang et al., 2021; Wang et al., 2021; Xie et al., 2021). In particular, BSRGAN (Zhang et al., 2021) and Real-ESRGAN (Wang et al., 2021) utilize GANs with practical degradation settings. Inspired by recent advanced diffusion models, StableSR (Wang et al., 2023) and DiffBIR (Lin et al., 2023) propose to exploit diffusion priors to generate realistic outputs. Although pretrained diffusion weights are well utilized in these methods, they do not make use of the semantic information embedded in vision-language models.

**Vision-language models (VLMs)** Recent works have demonstrated the great potential of applying pretrained VLMs to improve downstream tasks with generic visual and text representations (Radford et al., 2021; Jia et al., 2021; Li et al., 2022b). A classic VLM usually consists of a text encoder and an image encoder and tries to learn aligned multimodal features from noisy image-text pairs with contrastive learning (Radford et al., 2021). BLIP (Li et al., 2022b) further proposes to remove noisy web data by bootstrapping synthetic captions. Although VLMs provide a strong capability of zero-shot and label-free classification for downstream tasks, they have so far had limited effect on image restoration due to the need for specialized and accurate terminology. A noteworthy approach for fine-tuning vision-language models is so-called *prompt learning* (Zhou et al., 2022), where the prompt's context words are represented by learnable vectors that are then optimized for the downstream task.

**Text-to-image generation** Text-to-image models such as stable diffusion (Rombach et al., 2022) have gained extraordinary attention from both researchers and the general public. ControlNet (Zhang & Agrawala, 2023) builds upon this work and proposes to add controls to the diffusion network to make it adapt to task-specific conditions. InstructPix2Pix (Brooks et al., 2023) further combines GPT-3 (Brown et al., 2020) with stable diffusion to perform an instruction-based image-to-image translation. Although it can generate highly realistic images, it can not be directly applied to image restoration tasks since the latter requires highly accurate reconstruction abilities.

## 3 DEGRADATION-AWARE CLIP

At the core of our approach is the idea of controlling a pre-trained CLIP model to output the high-quality image feature from a corrupted image while simultaneously predicting the degradation type.

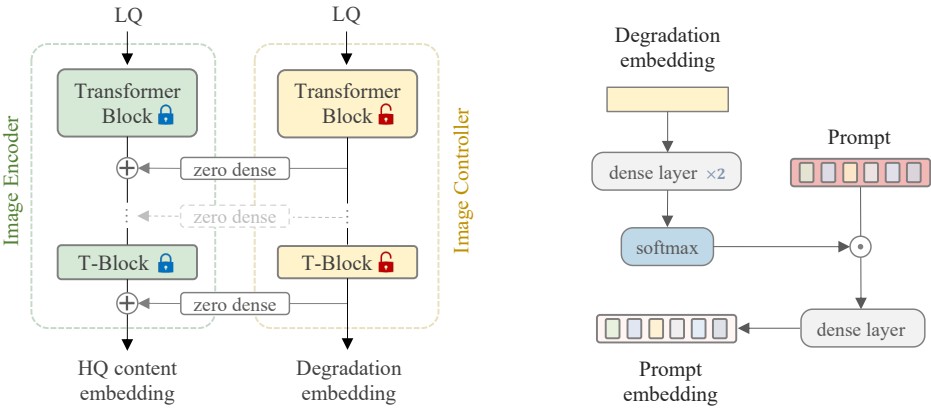

(a) Controller for ViT-based image encoder    (b) Prompt with degradation embeddings

Figure 3: (a) Controlling the ViT-based image encoder with a trainable controller. (b) Learning the prompt with degradation embeddings predicted from the controller.

As summarised in Figure 2, the image content embedding $e_c^I$ matches the clean caption embedding $e_c^T$. Moreover, the image degradation embedding $e_d^I$ predicted by the controller specifies the corruption type of the input, i.e. the corresponding degradation embedding $e_d^T$ from text encoder. These features can then be integrated into other image restoration models to improve their performance.

## 3.1 IMAGE CONTROLLER

The image controller is a copy of the CLIP image encoder but wrapped with a few zero-initialised connections to add controls to the encoder. It manipulates the outputs of all encoder blocks to control the prediction of the image encoder. In this paper, we use ViT (Dosovitskiy et al., 2020) as the default backbone for both the encoder and the controller. Figure 3(a) illustrates the controlling procedure, where the output of the controller consists of two parts: an image degradation embedding $e_d^I$ and hidden controls $h_c$. Note that the latter contains all outputs from the transformer blocks, which are subsequently added to the corresponding encoder blocks to control their predictions. The connections between the transformer blocks are simple dense neural networks with all parameters initialized to zero, which gradually influence the image encoder during training (Zhang & Agrawala, 2023). Since our training dataset is tiny compared to the web-scale datasets used in VLMs, this controlling strategy alleviates overfitting while preserving the capability of the original image encoder.

**Optimising the image controller** We freeze all weights of the pretrained CLIP model and only fine-tune the image controller. To make the degradation-embedding spaces discriminative and well-separated, we use a contrastive objective (Tian et al., 2020) to learn the embedding matching process. Let $N$ denote the number of paired embeddings (from text encoder and image encoder/controller) in a training batch. The contrastive loss is defined as:

$$\mathcal{L}_{\text{con}}(\boldsymbol{x}, \boldsymbol{y}) = -\frac{1}{N} \sum_{i=1}^{N} \log \left( \frac{\exp\left(\boldsymbol{x}_i^\mathsf{T} \boldsymbol{y}_i / \tau\right)}{\sum_{j=1}^{N} \exp(\boldsymbol{x}_i^\mathsf{T} \boldsymbol{y}_j / \tau)} \right), \tag{1}$$

where $\boldsymbol{x}$ and $\boldsymbol{y}$ are normalised vectors, and $\tau$ is a learnable temperature parameter that controls the contrastive strength. Minimising Equation 1 amounts to optimising the cosine similarity between correctly paired embeddings while enlarging the difference with other embeddings, similar to the cross-entropy loss (Radford et al., 2021). To optimise both content and degradation embeddings, we use the following joint objective:

$$\mathcal{L}_c(\omega) = \mathcal{L}_{\text{con}}(e_c^I, e_c^T; \omega) + \mathcal{L}_{\text{con}}(e_d^I, e_d^T; \omega), \tag{2}$$

where $\omega$ represents the learnable parameters of the controller. Note that all image-based embeddings (i.e., $e_c^I$ and $e_d^I$) are obtained from the LQ image and all text-based embeddings (i.e., $e_c^T$ and $e_d^T$) are from the clean caption and real degradation. Learning to align these embeddings enables DA-CLIP to predict real degradation types and HQ content features for corrupted image inputs.

## 3.2 IMAGE RESTORATION WITH DA-CLIP

We use IR-SDE (Luo et al., 2023a) as the base framework for image restoration. It adapts a U-Net architecture similar to DDPM (Ho et al., 2020) but removes all self-attention layers. To inject clean content embeddings into the diffusion process, we introduce a cross-attention (Rombach et al., 2022) mechanism to learn semantic guidance from pre-trained VLMs. Considering the varying input sizes in image restoration tasks and the increasing cost of applying attention to high-resolution features, we only use cross-attention in the bottom blocks of the U-Net for sample efficiency.

On the other hand, the predicted degradation embeddings are useful for unified image restoration, where the aim is to process low-quality images of multiple degradation types with a single model (Li et al., 2022a). As illustrated in Figure 1, our DA-CLIP accurately classifies the degradation across different datasets and various degradation types, which is crucial for unified image restoration (Li et al., 2022a). Moreover, to make use of these degradation embeddings, we combine them with a prompt learning (Zhou et al., 2022) module to further improve the results, as shown in Figure 3(b). Given state $x_t$ and low-quality image $\mu$, our final diffusion network is conditioned on the time $t$ and additional embeddings $e_c^I$ and $e_d^I$, as $\epsilon_\theta(x_t, \mu, t, e_c^I, e_d^I)$, which can be trained with either noise-matching loss (Ho et al., 2020) or maximum likelihood loss (Luo et al., 2023a;b).

Generally, we can use cross-attention to integrate content embedding into networks to improve their performance on an image restoration task. In contrast, the prompt module combined with degradation embedding specifically aims to improve the classification of the degradation type in the context of unified image restoration.

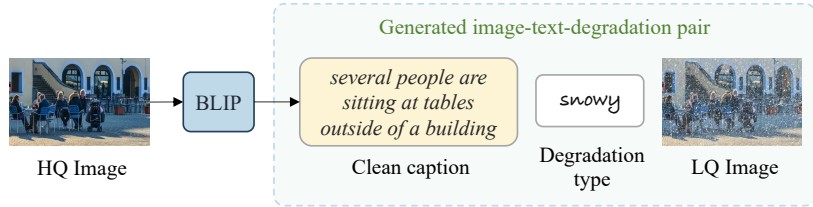

Figure 4: An example of generating the image-text-degradation tuple with BLIP. The clean caption generated from the HQ image is accurate and does not convey the degradation information.

## 4 DATASET CONSTRUCTION

A crucial point of this paper is to leverage powerful vision-language models to learn multiple image degradations and extract degradation-free features for image restoration. For this purpose, we collect a large dataset with ten different image degradation types: *blurry*, *hazy*, *JPEG-compression*, *low-light*, *noisy*, *raindrop*, *rainy*, *shadowed*, *snowy*, and *inpainting*. Table 1 summarises the tasks and the number of training and testing images for each degradation type, and more details are provided in Appendix A. Example low-quality images for the ten degradations are also shown in Figure 1.

**Generating image-text-degradation pairs** In order to train DA-CLIP on the mixed-degradation dataset, we use the bootstrapped vision-language framework BLIP (Li et al., 2022b) to generate synthetic captions for all HQ images. Since the inputs are clean, the generated captions are assumed to be accurate and of high-quality. As illustrated in Figure 4, we are then able to construct image-text-degradation pairs by directly combining these clean captions, LQ images, and the corresponding degradation types. This dataset allows us to train either vision-language models (based on image-text-degradation pairs) or a unified image restoration framework (based on LQ-HQ image pairs).

Table 1: Details of the collected training and testing datasets with different image degradation types.

| Dataset | Blurry | Hazy | JPEG | Low-light | Noisy | Raindrop | Rainy | Shadowed | Snowy | Inpainting |
|---------|--------|------|------|-----------|-------|----------|-------|----------|-------|------------|
| #Train | 2 103 | 6 000 | 3 550 | 485 | 3 550 | 861 | 1 800 | 2 680 | 1 872 | 29 900 |
| #Test | 1 111 | 1 000 | 29 | 15 | 68 | 58 | 100 | 408 | 601 | 100 |

## 5 EXPERIMENTS

We experimentally evaluate our method on two types of tasks: *degradation-specific* image restoration and *unified* image restoration. In the degradation-specific setting (Section 5.1), restoration models are separately trained for each of the considered degradation types. In unified image restoration (Section 5.2), a single model is instead jointly trained on all degradation types. Implementation details and additional results are provided in Appendix B and Appendix C.

**Model Evaluation** Our DA-CLIP is mainly evaluated in terms of how it affects the performance of the downstream image restoration model, which we evaluate both on multiple degradation-specific tasks and on unified image restoration. IR-SDE (Luo et al., 2023a) is used as our primary baseline model. We use the Learned Perceptual Image Patch Similarity (LPIPS) (Zhang et al., 2018) and Fréchet inception distance (FID) (Heusel et al., 2017) as our main metrics for perceptual evaluation, but also report PSNR and SSIM (Wang et al., 2004) for reference. Additionally, we evaluate DA-CLIP in terms of how well it can classify the ten different degradation types on the mixed dataset.

### 5.1 DEGRADATION-SPECIFIC IMAGE RESTORATION

We integrate our DA-CLIP into the baseline diffusion model IR-SDE and evaluate them on four degradation-specific tasks: image deraining on the Rain100H dataset (Yang et al., 2017), low-light image enhancement on LOL (Wei et al., 2018), image deblurring on the GoPro dataset (Nah et al., 2017), and image dehazing on RESIDE-6k (Qin et al., 2020). All training and testing datasets are taken from the mixed degradation dataset described in Section 4.

**Comparison approaches** For all tasks, we compare our method to the prevailing approaches in their respective fields such as 1) JORDER (Yang et al., 2019), PReNet (Ren et al., 2019), and MPR-Net (Zamir et al., 2021) for deraining; 2) EnlightenGAN (Jiang et al., 2021), MIRNet (Zamir et al., 2020), and URetinex-Net (Wu et al., 2022) for low-light enhancement; 3) DeepDeblur (Nah et al., 2017), DeblurGAN (Kupyn et al., 2018), and DeblurGAN-v2 (Kupyn et al., 2019) for GoPro deblurring; 4) GCANet (Chen et al., 2019), GridDehazeNet (Liu et al., 2019), and DehazeFormer (Song et al., 2023) for image dehazing. We also compare with MAXIM (Tu et al., 2022), an advanced network architecture that achieves state-of-the-art performance on multiple degradation-specific tasks.

**Results** The quantitative results on different datasets are summarised in Table 2. Our method achieves the best perceptual results across all tasks, and even sets a new state-of-the-art performance for all metrics on the image deraining task. Compared to the baseline method IR-SDE, our approach consistently improves its results for all datasets and metrics, demonstrating that the HQ content embedding from DA-CLIP leads to better performance for downstream image restoration models. A visual comparison of our method with other approaches is also illustrated in Figure 5. Our method produces mostly clear and visually appealing results that are close to the HQ images.

### 5.2 UNIFIED IMAGE RESTORATION

We evaluate our method for unified image restoration on the mixed degradation dataset which contains ten different degradation types (see Section 4 for details).

**Comparison approaches** We compare our method with three approaches: Restormer (Zamir et al., 2022), AirNet (Li et al., 2022a), and PromptIR (Potlapalli et al., 2023). Restormer is an advanced transformer-based network. AirNet trains an extra encoder to differentiate degradation types using contrastive learning, whereas PromptIR employs a visual prompt module to guide the restoration. The latter two methods are designed specifically for unified image restoration.

**Results** We illustrate the comprehensive comparisons in Figure 6 and also provide the average results across all ten degradation types in Table 3. The results show that our method achieves the best perceptual results (especially in terms of FID) across the ten degradations, while still having a good distortion performance. More importantly, by simply integrating DA-CLIP into the network, we significantly outperform the IR-SDE baseline in terms of all four metrics. A visual comparison is shown in Figure 8. On JPEG and noise removal, our method produces realistic-looking but somewhat noisy images, giving lower distortion metrics for those particular tasks. In Appendix B.3, we evaluate methods on out-of-distribution light rain images, in which DA-CLIP surpasses all other unified approaches, demonstrating that our method can generalize to unseen degradation levels.

Table 2: Quantitative comparison between our method with other state-of-the-art approaches on four different *degradation-specific* datasets. The best results are marked in boldface.

| Method | Distortion | | Perceptual | |
|---|---|---|---|---|
| | PSNR↑ | SSIM↑ | LPIPS↓ | FID↓ |
| JORDER | 26.25 | 0.835 | 0.197 | 94.58 |
| PReNet | 29.46 | 0.899 | 0.128 | 52.67 |
| MPRNet | 30.41 | 0.891 | 0.158 | 61.59 |
| MAXIM | 30.81 | 0.903 | 0.133 | 58.72 |
| IR-SDE | 31.65 | 0.904 | 0.047 | 18.64 |
| Ours | **33.91** | **0.926** | **0.031** | **11.79** |

(a) Deraining results on the Rain100H dataset.

| Method | Distortion | | Perceptual | |
|---|---|---|---|---|
| | PSNR↑ | SSIM↑ | LPIPS↓ | FID↓ |
| EnlightenGAN | 17.61 | 0.653 | 0.372 | 94.71 |
| MIRNet | **24.14** | 0.830 | 0.250 | 69.18 |
| URetinex-Net | 19.84 | 0.824 | 0.237 | 52.38 |
| MAXIM | 23.43 | **0.863** | 0.098 | 48.59 |
| IR-SDE | 20.45 | 0.787 | 0.129 | 47.28 |
| Ours | 23.77 | 0.830 | **0.083** | **34.03** |

(b) Low-light enhancement on the LOL dataset.

| Method | Distortion | | Perceptual | |
|---|---|---|---|---|
| | PSNR↑ | SSIM↑ | LPIPS↓ | FID↓ |
| DeepDeblur | 29.08 | 0.913 | 0.135 | 15.14 |
| DeblurGAN | 28.70 | 0.858 | 0.178 | 27.02 |
| DeblurGANv2 | 29.55 | 0.934 | 0.117 | 13.40 |
| MAXIM | **32.86** | **0.940** | 0.089 | 11.57 |
| IR-SDE | 30.70 | 0.901 | 0.064 | 6.32 |
| Ours | 30.88 | 0.903 | **0.058** | **6.15** |

(c) Deblurring results on the GoPro dataset.

| Method | Distortion | | Perceptual | |
|---|---|---|---|---|
| | PSNR↑ | SSIM↑ | LPIPS↓ | FID↓ |
| GCANet | 26.59 | 0.935 | 0.052 | 11.52 |
| GridDehazeNet | 25.86 | 0.944 | 0.048 | 10.62 |
| DehazeFormer | **30.29** | **0.964** | 0.045 | 7.58 |
| MAXIM | 29.12 | 0.932 | 0.043 | 8.12 |
| IR-SDE | 25.25 | 0.906 | 0.060 | 8.33 |
| Ours | 30.16 | 0.936 | **0.030** | **5.52** |

(d) Dehazing results on the RESIDE-6k dataset.

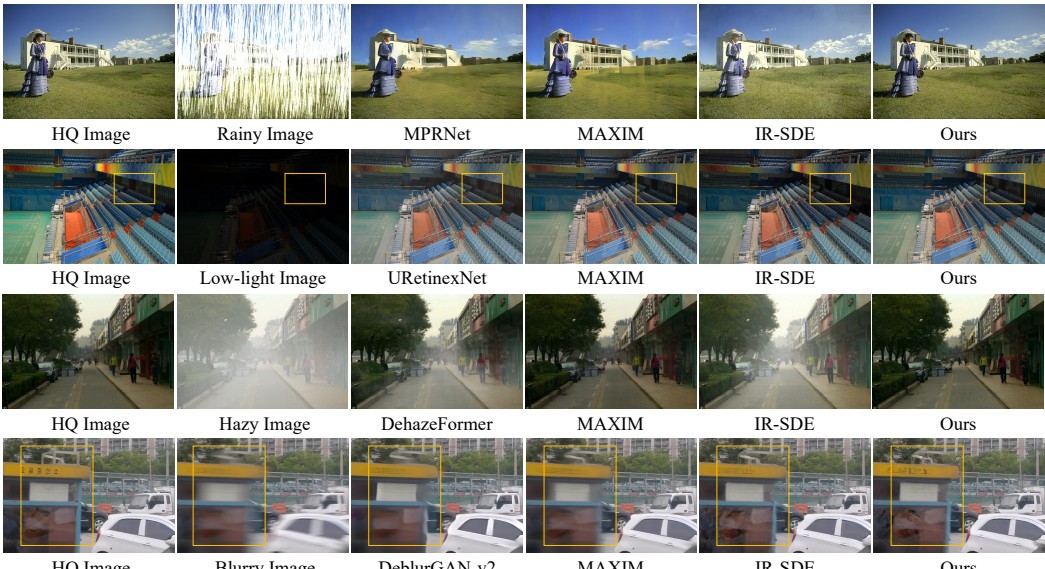

Figure 5: Comparison of our method with other approaches on 4 different *degradation-specific* tasks.

Moreover, we further integrate[1] DA-CLIP into an MSE-based network NAFNet (Chen et al., 2022) as a variant of our method. The results are illustrated in Table 3 and Figure 7. We observe that adding our degradation context significantly improves the results, and the final performance of NAFNet with DA-CLIP even surpasses PromptIR across all metrics. This demonstrates that DA-CLIP can be integrated with both diffusion-based and direct restoration models, further improving their performance for unified image restoration. More results can be found in Appendix B.4.

Finally, we evaluate DA-CLIP in terms of degradation type classification as shown in Figure 1 and Table 4 (in the Appendix). The original CLIP model achieves less than 2% accuracy in recognizing noisy and raindrop images, and even 0% accuracy for inpainting, which would confuse a downstream unified image restoration model. By training the controller on the mixed degradation dataset, DA-CLIP perfectly predicts all degradations except blurry, for which it achieves 91.6% accuracy.

---

[1]We add the prompt module to all blocks and use cross-attention in the bottom layers.

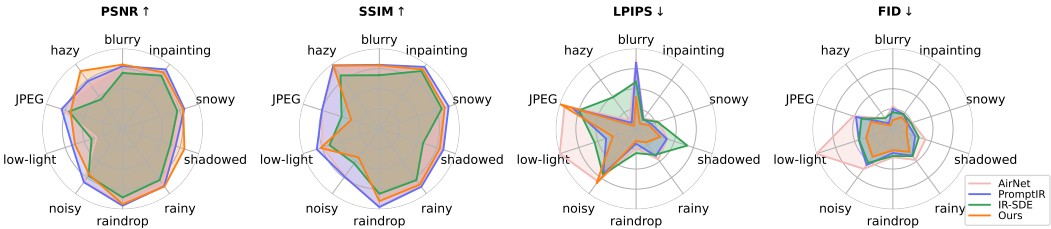

Figure 6: Comparison of our method with AirNet, PromptIR and IR-SDE for *unified* image restoration. Each radar plot reports results for the ten different degradation types, for one particular metric. For the perceptual metrics LPIPS and FID (the two rightmost plots), a lower value is better.

Table 3: Comparison of the average results over ten different datasets on the *unified* image restoration task.

| Method | Distortion | | Perceptual | |
|---|---|---|---|---|
| | PSNR↑ | SSIM↑ | LPIPS↓ | FID↓ |
| NAFNet | 26.34 | 0.847 | 0.159 | 55.68 |
| NAFNet + Degradation | 27.02 | 0.856 | 0.146 | 48.27 |
| NAFNet + DA-CLIP | **27.22** | **0.861** | 0.145 | 47.94 |
| Restormer | 26.43 | 0.850 | 0.157 | 54.03 |
| AirNet | 25.62 | 0.844 | 0.182 | 64.86 |
| PromptIR | 27.14 | 0.859 | 0.147 | 48.26 |
| IR-SDE | 23.64 | 0.754 | 0.167 | 49.18 |
| Ours | 27.01 | 0.794 | **0.127** | **34.89** |

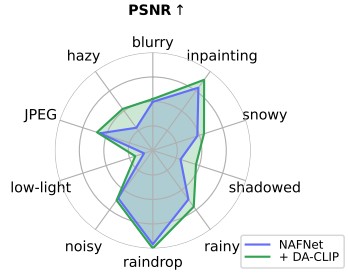

Figure 7: NAFNet with DA-CLIP for *unified* image restoration.

## 5.3 DISCUSSION AND ANALYSIS

We have two different embeddings from the DA-CLIP: an *HQ content* embedding and a *degradation* embedding, which are both predicted from the LQ image. The former can be used for improving general image restoration performance, whereas the latter is predicted to classify degradation types for unified image restoration models. As can be observed in Figure 9a, separately applying either the degradation embedding or the HQ content embedding improves the unified restoration performance, and the performance is further improved by applying both embeddings as in our DA-CLIP. Moreover, directly embedding the ground truth HQ image and the real degradation type leads to an upper-bound performance of our method, as illustrated in Figure 9b.

We also compare using the HQ content image embedding from our DA-CLIP with the original image content embedding obtained from the OpenAI CLIP (Radford et al., 2021). For the unified image restoration task, we observe in Figure 9c that using the original CLIP content embedding fails to substantially improve the performance. For the degradation-specific setting as shown in Figure 9d, we also observe that our DA-CLIP clearly outperforms the baseline of using the original CLIP.

As an alternative to using the HQ content and degradation embeddings of DA-CLIP (from the LQ image), the degradation type and HQ caption could be encoded directly using the CLIP text encoder. This alternative approach thus requires access to the ground truth degradation label and caption text for each LQ image. Surprisingly, we observe in Figure 9e that the two degradation embeddings give very similar performance. And Figure 9f further shows that caption embeddings are also helpful for image restoration. Moreover, while using the ground truth text embeddings improves the baseline performance, it is slightly inferior compared to using our DA-CLIP image embeddings (Figure 9g), meaning that DA-CLIP has learned to accurately predict the degradation type and HQ content embeddings. Finally, an analysis of the degradation prompt module is provided in Figure 9h, demonstrating that prompt learning also facilitates unified image restoration.

While we have demonstrated the effectiveness of our proposed DA-CLIP in various settings, it is worth acknowledging one potential limitation: increased model complexity and computational cost. As can be observed in Table 8 in the Appendix, DA-CLIP significantly increases the memory requirements compared to the baseline models (both NAFNet and IR-SDE). The test-time computational cost (FLOPs and runtime) is however virtually unaffected.

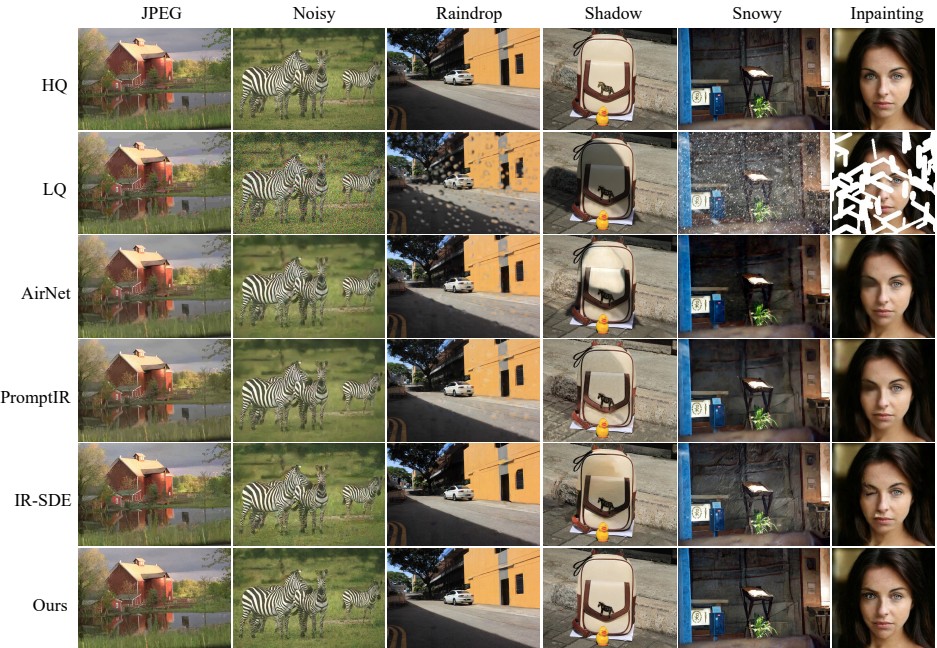

Figure 8: Comparison of our method with other approaches on the *unified* image restoration. All results in each row are produced by sending images with different degradations to the same model.

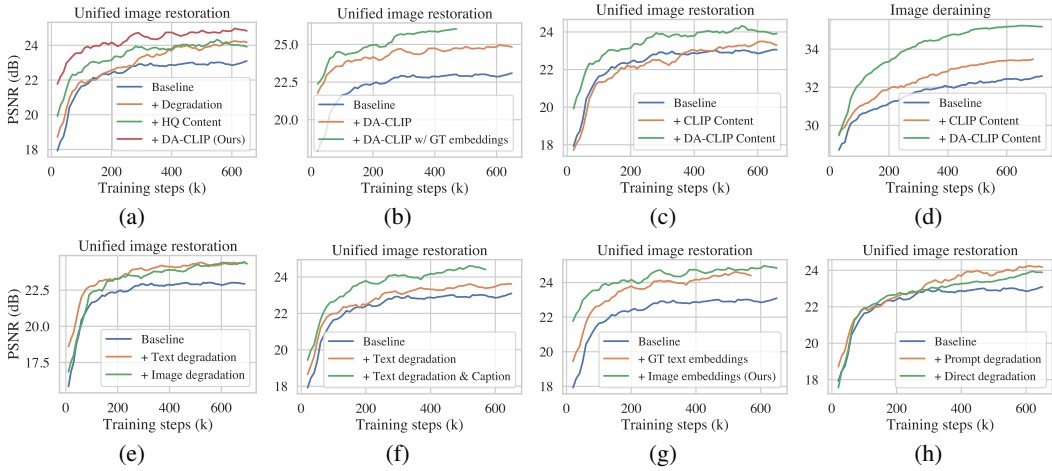

Figure 9: Training curves of model variations, demonstrating the effectiveness of our DA-CLIP.

# 6    CONCLUSION

This paper presents DA-CLIP to leverage large-scale pretrained vision-language models as a universal framework for image restoration. At the core of our approach is a controller that accurately predicts the degradation embeddings from LQ images and also controls the CLIP image encoder to output clean content embeddings. To train DA-CLIP, we construct a mixed degradation dataset containing synthetic captions from HQ images. DA-CLIP is then integrated into downstream image restoration models using a prompt learning module and a cross-attention mechanism. Experimental evaluation on both degradation-specific and unified image restoration tasks demonstrates that DA-CLIP consistently improves the restoration performance, across a variety of degradation types. On the other hand, we notice that the current dataset makes it hard to restore multiple degradations in the same scene. In future work, we are interested in creating practical models that are more robust to real-world captured photos and are able to fully restore images with mixed degradation types.

ACKNOWLEDGEMENTS

This research was supported by the *Wallenberg AI, Autonomous Systems and Software Program (WASP)* funded by the Knut and Alice Wallenberg Foundation; by the project *Deep Probabilistic Regression – New Models and Learning Algorithms* (contract number: 2021-04301) funded by the Swedish Research Council; and by the *Kjell & Märta Beijer Foundation*. The computations were enabled by the *Berzelius* resource provided by the Knut and Alice Wallenberg Foundation at the National Supercomputer Centre.

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

# A  MORE DETAILS ABOUT DATASETS

In this section, we give more details about the mixed degradation dataset in Section 4. We collect images for 10 different image restoration tasks, including *blurry*, *hazy*, *JPEG-compressing*, *low-light*, *noisy*, *raindrop*, *rainy*, *shadowed*, *snowy*, and *inpainting*, as shown in Figure 1. The details of these datasets are listed below:

- *Blurry*: collected from the GoPro (Nah et al., 2017) dataset containing 2103 and 1111 training and testing images, respectively.
- *Hazy*: collected from the RESIDE-6k (Qin et al., 2020) dataset which has mixed indoor and outdoor images with 6000 images for training and 1000 images for testing.
- *JPEG-compressing*: the training dataset has 3440 images collected from DIV2K (Agusts-son & Timofte, 2017) and Flickr2K (Timofte et al., 2017). The testing dataset contains 29 images from LIVE1 (Sheikh, 2005). Moreover, all LQ images are synthetic data with a JPEG quality factor of 10.
- *Low-light*: collected from the LOL (Wei et al., 2018) dataset containing 485 images for training and 15 images for testing.
- *Noisy*: the training dataset is the same as that in *JPEG-compressing* but all LQ images are generated by adding Gaussian noise with noise level 50. The testing images are from CBSD68 (Martin et al., 2001) and also added that Gaussian noise.
- *Raindrop*: collected from the RainDrop (Qian et al., 2018) dataset containing 861 images for training and 58 images for testing.
- *Rainy*: collected from the Rain100H (Yang et al., 2017) dataset containing 1800 images for training and 100 images for testing.
- *Shadowed*: collection from the SRD (Qu et al., 2017) dataset containing 2680 images for training and 408 images for testing.
- *Snowy*: collected from the Snow100K-L (Liu et al., 2018) dataset. Since the original dataset is too large (100K images), we only use a subset which contains 1872 images for training and 601 images for testing.
- *Inpainting*: we use CelebaHQ as the training dataset and divide 100 images with 100 thin masks from RePaint (Lugmayr et al., 2022) for testing.

We also provide several visual examples for each task for a better understanding of the 10 degradations and datasets, as shown in Figure 10.

# B  IMPLEMENTATION DETAILS AND MORE ANALYSIS

## B.1  IMPLEMENTATION DETAILS

The base CLIP model uses *ViT-B-32* as the backbone for its image encoder, with weights pretrained on the LAION-2B dataset (Schuhmann et al., 2022). Built upon it, we fine-tune the DA-CLIP on the mixed degradation dataset with a batch size of $3\,136$ ($784 \times 4$) and learning rate $3 \times 10^{-5}$. In preprocessing, all inputs are normalized in the range $[0, 1]$ and resized to $224 \times 224$ with bicubic interpolation. We train the DA-CLIP model on four NVIDIA A100 GPUs for 50 epochs, in approximately 3 hours. For the restoration model, we use a batch size of 16 and randomly crop images to $256 \times 256$ for data augmentation. The initial learning rate is $2 \times 10^{-4}$. We use the AdamW (Loshchilov & Hutter, 2017) optimizer ($\beta_1 = 0.9, \beta_1 = 0.99$) with cosine decay for a total of 700K iterations. All training is done using one A100 GPU for about 5 days.

## B.2  ADDITIONAL ANALYSIS OF DA-CLIP

We provide the degradation classification results of three strategies: 1) retraining CLIP on our dataset; 2) fine-tuning CLIP from pretrained weights; and 3) adding a controller but without zero-initializing its dense layers. The prompt is set to "a [*degradation type*] photo" for all models. The results are reported in Table 4. Obviously, CLIP with finetuning significantly improves the results of

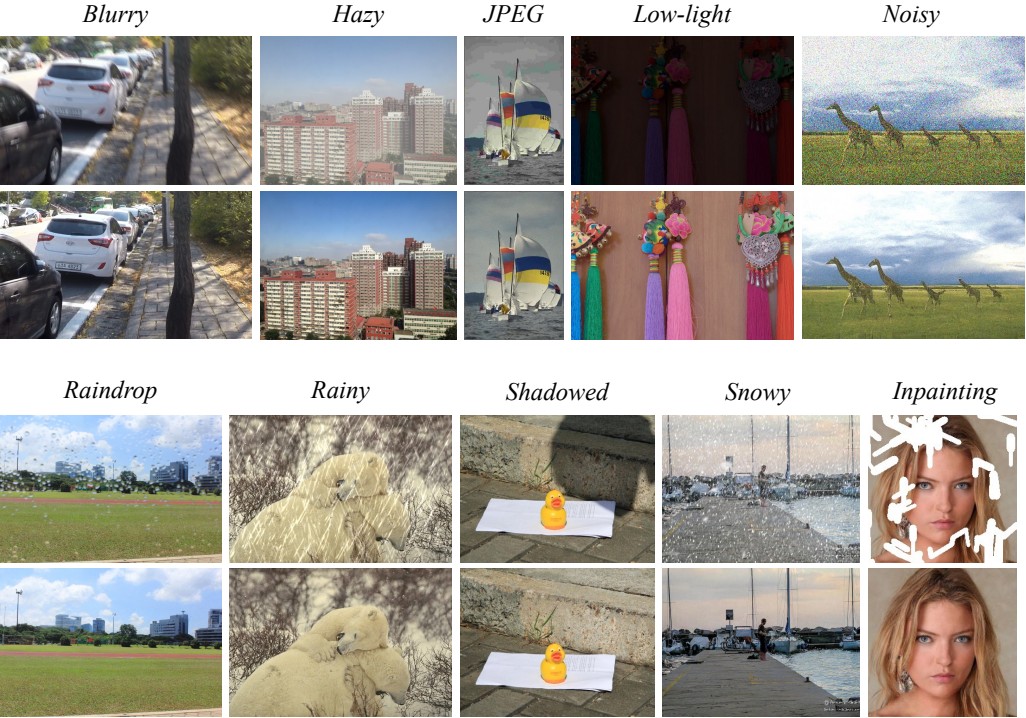

| Blurry | Hazy | JPEG | Low-light | Noisy |
| --- | --- | --- | --- | --- |

| Raindrop | Rainy | Shadowed | Snowy | Inpainting |
| --- | --- | --- | --- | --- |

Figure 10: Example images with 10 image restoration tasks. For each task, the first row is the corrupted input and the second row is the result produced by our *unified* image restoration model.

Table 4: Accuracies of the degradation classification. The average accuracy for each method is provided in the last column. 'DA-CLIP w/o Zero' means adding a controller without zero-initialising its dense layers.

| Model | Blurry | Hazy | JPEG | Low-light | Noisy | Raindrop | Rainy | Shadowed | Snowy | Inpainting | Average |
| --- | --- | --- | --- | --- | --- | --- | --- | --- | --- | --- | --- |
| OpenAI CLIP | 80.3 | 72.9 | 20.7 | 13.3 | 1.5 | 1.7 | 80 | 29.9 | 27 | 0 | 26.7 |
| Retrain CLIP | 65.3 | 97.8 | 65.5 | 100 | 100 | 86.2 | 99 | 94.6 | 88 | 100 | 89.6 |
| CLIP + finetune | 95.6 | 100 | 100 | 100 | 100 | 100 | 100 | 100 | 100 | 100 | 99.6 |
| DA-CLIP w/o Zero | 90.1 | 99 | 100 | 100 | 97.1 | 96.5 | 100 | 99.7 | 99 | 100 | 98.1 |
| DA-CLIP (Ours) | 91.6 | 100 | 100 | 100 | 100 | 100 | 100 | 100 | 100 | 100 | 99.2 |

direct retraining, demonstrating the effectiveness of large-scale vision-language model pretraining. Note that although fine-tuning CLIP achieves slightly better performance on blurry classification than DA-CLIP, it is not able to predict HQ content embeddings from the LQ image input, which leads to limited effects on downstream restoration models. Moreover, initializing dense layers to zero can further improve the accuracy of all datasets without adding additional costs.

Table 5: Evaluating unified models on the Rain100L dataset for OOD analysis.

| Method | Distortion | | Perceptual | |
| --- | --- | --- | --- | --- |
| | PSNR↑ | SSIM↑ | LPIPS↓ | FID↓ |
| AirNet | 30.07 | 0.935 | 0.114 | 45.56 |
| PromptIR | 32.77 | 0.950 | 0.081 | 43.31 |
| IR-SDE | 30.13 | 0.904 | 0.098 | 35.05 |
| Ours | **36.61** | **0.967** | **0.025** | **11.98** |

Table 6: Ablation experiments of applying DA-CLIP to other MSE-based methods on *unified* image restoration. PromptIR is built upon Restormer but with additional degradation prompt modules.

| Method | PSNR↑ | SSIM↑ | LPIPS↓ | FID↓ |
|---|---|---|---|---|
| NAFNet | 26.34 | 0.847 | 0.159 | 55.68 |
| + Degradation embedding | 27.02 | 0.856 | 0.146 | 48.27 |
| + Degradation and Content embedding | 27.22 | 0.861 | 0.145 | 47.94 |
| Restormer | 26.43 | 0.850 | 0.157 | 54.03 |
| PromptIR (Restormer + degradation) | 27.14 | 0.859 | 0.147 | 48.26 |
| PromptIR + Content embedding | 27.26 | 0.861 | 0.145 | 47.75 |

Table 7: Results of integrating the DA-CLIP into NAFNet for degradation-specific tasks.

| Method | Distortion | | Perceptual | |
|---|---|---|---|---|
| | PSNR↑ | SSIM↑ | LPIPS↓ | FID↓ |
| NAFNet | 31.49 | 0.903 | 0.087 | 31.05 |
| NAFNet+DA-CLIP | 31.68 | 0.907 | 0.086 | 31.02 |

(a) Deraining results on the Rain100H dataset.

| Method | Distortion | | Perceptual | |
|---|---|---|---|---|
| | PSNR↑ | SSIM↑ | LPIPS↓ | FID↓ |
| NAFNet | 23.09 | 0.839 | 0.122 | 57.45 |
| NAFNet+DA-CLIP | 23.72 | 0.844 | 0.116 | 50.69 |

(b) Low-light enhancement on the LOL dataset.

## B.3   EXPERIMENTS ON OUT-OF-DISTRIBUTION (OOD) DEGRADATION

To evaluate our model's generalization capability, we apply our model on an additional light rain dataset Rain100L Yang et al. (2017). Since our mixed-degradation dataset only contains heavy rain images (from the Rain100H dataset), Rain100L is an out-of-distribution dataset with a relatively low-level degradation. Table 5 summarises the results. It is observed that our method performs surprisingly well on this unseen dataset and surpasses all other unified approaches by a significant margin. In addition, we also provide some real-world visual examples in Figure 17, showing that our model seems to generalize well compared to the AirNet and PrompIR baselines.

## B.4   IMPROVING MSE-BASED METHOD WITH DA-CLIP

It is worth noting that our DA-CLIP can not only benefit the diffusion-based image restoration approach but also facilitate standard MSE-based model learning. Table 6 provides two ablation experiments on NAFNet (Chen et al., 2022) and Restormer (Zamir et al., 2022). Note that PromptIR is built upon Restormer but with additional degradation prompt modules. As can be seen, the degradation context is useful for both methods, and adding additional content embedding can further improve their performance.

## B.5   TRAINING CURVES ON SINGLE DEGRADATION TASKS

To further illustrate the effectiveness of our method, we compare the training curves between the baseline model IR-SDE (Luo et al., 2023a) and that with our DA-CLIP embeddings on four different degradation-specific image restoration tasks: image deraining, low-light enhancement, image deblurring, and image dehazing. The results are shown in Figure 11, from which we can see the training with our DA-CLIP obviously performs better than the baseline model.

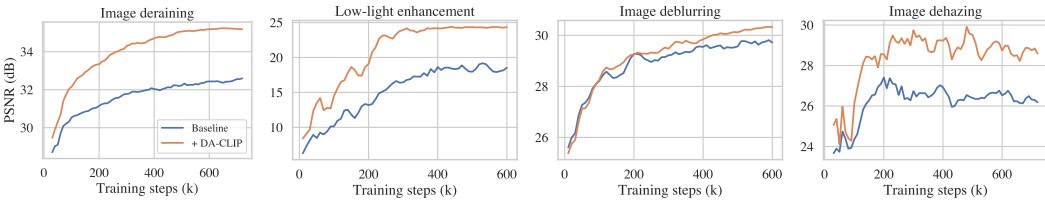

Figure 11: Ablation studies on the *degradation-specific* image restoration task.

### B.6 ANALYSIS ABOUT THE MODEL COMPLEXITY

We have shown the effectiveness of applying our DA-CLIP to various image restoration models and tasks. However, it is worth acknowledging one potential limitation: the model complexity is also increased along with the DA-CLIP inference and with additional prompt modules and cross-attention layers. The comparison of model complexities is shown in Table 8. DA-CLIP significantly increases the memory requirements compared to the baseline models (both NAFNet and IR-SDE). The test-time computational cost (FLOPs and runtime) is however virtually unaffected.

### B.7 ANALYSIS OF PATCH SIZES

Intuitively, large patch sizes always contain more semantic information that might be important for guiding image restoration. We explore this property by training our model with two different patch sizes on the unified image restoration task. As shown in Figure 12, increasing the patch size from $128 \times 128$ to $256 \times 256$ improves the training process, which is consistent with our conjecture.

Table 8: Comparison of the number of parameters, model computational efficiency, and inference time. The flops and inference time are computed on face inpainting images of size $256 \times 256$. *Note that MAXIM is implemented with the JAX GPU version.*

| METHOD | MAXIM | PROMPTIR | NAFNET | NAFNET + DA-CLIP | IR-SDE | IR-SDE + DA-CLIP |
|---|---|---|---|---|---|---|
| #PARAM | 14.1M | 33M | 67.8M | 86.1M + 125.2M | 36.2M | 48.9M + 125.2M |
| FLOPS | 216G | 158G | 63G | 65G + 0.5G | 117G | 118G + 0.5G |
| RUNTIME | 2.9s | 0.1s | 0.08s | 0.08s | 4.2s | 4.53s + 0.06s |

## C ADDITION EXPERIMENTAL RESULTS

In this section, we provide more details and additional experimental results.

### C.1 DETAILED QUANTITATIVE RESULTS FOR UNIFIED IMAGE RESTORATION

We provide some examples to show the degradation prediction on specific images with different corruptions in Figure 13. In Section 5.2 we have reported the average results and radar figures for different metrics. Here we also provide a more detailed comparison between our method with other approaches for unified image restoration. The results on four metrics are shown in Table 9, Table 10, Table 11, and Table 12. Additionally, we also provide a detailed comparison between NAFNet (Chen et al., 2022) and its variant (with DA-CLIP) in all tables, which illustrates the potential of applying our DA-CLIP to other general image restoration frameworks.

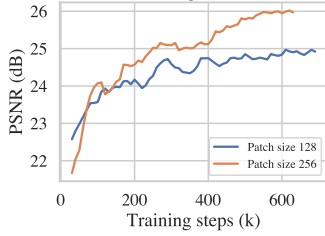

Figure 12: Comparison of training curves with different patch sizes.

### C.2 ADDITIONAL VISUAL RESULTS

The additional visual results for unified image restoration are shown in Figure 14, Figure 16. Figure 15 shows the comparison with Real-ESRGAN and StableSR on a compressed image and a blurry image. Figure 17 also illustrates some examples of testing our method on real-world images. We basically compare with PromptIR (Potlapalli et al., 2023) and IR-SDE (Luo et al., 2023a) but also add the results generated from AirNet (Li et al., 2022a).

## D LIMITATION ON MIXED DEGRADATIONS

Since the mixed degradation dataset contains a single degradation label for each image, our current model has not been trained to restore multiple degradations in the same scene. For example, the

raindrop image in Figure 8 contains a shadow area, but our model only removes the raindrop degradation. Also, due to this limitation, this paper can't well-explore and justify the effectiveness of linguistic components based on mixed degradations.

In future work, we are very interested in 1) creating practical models that are able to process real-world degradations and more complex degradations; 2) exploring other pretrained VLM backbones for more robust visual representations/embeddings; 3) exploring more interesting works for the linguistic side (from text encoder) such as instruction-based image restoration.

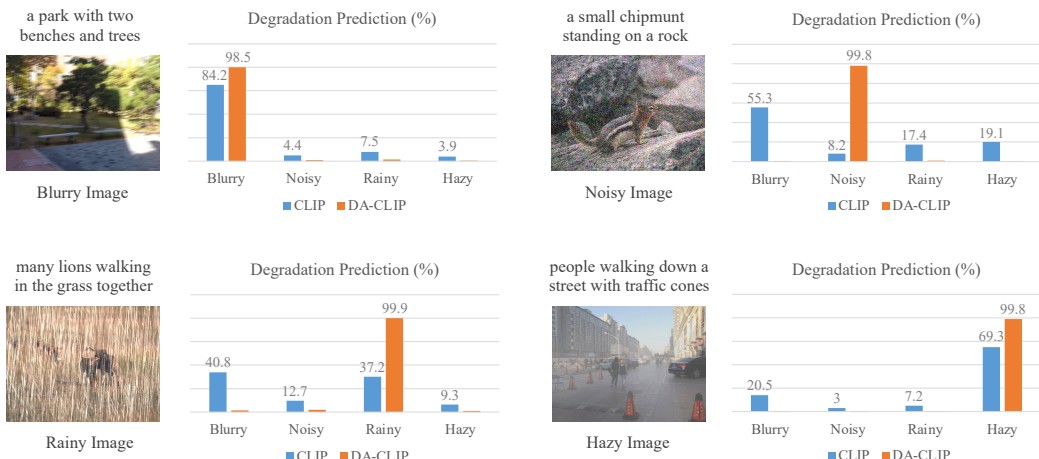

Figure 13: Comparison of CLIP and DA-CLIP for image degradation prediction. The captions above the corrupted images are generated using BLIP (Li et al., 2022b) with corresponding clean images.

Table 9: Comparison of our method with other unified image restoration approaches *on PSNR*.

| Model | Blurry | Hazy | JPEG | Low-light | Noisy | Raindrop | Rainy | Shadowed | Snowy | Inpainting | Average |
|---|---|---|---|---|---|---|---|---|---|---|---|
| NAFNet | 26.12 | 24.05 | 26.81 | 22.16 | 27.16 | 30.67 | 27.32 | 24.16 | 25.94 | 29.03 | 26.34 |
| NAFNet+DA-CLIP | 26.40 | 26.39 | **27.13** | 23.09 | 27.43 | 31.08 | 28.23 | 25.80 | 26.64 | 30.01 | **27.22** |
| Restormer | 26.34 | 23.75 | 26.90 | 22.17 | 27.25 | 30.85 | 27.91 | 23.33 | 25.98 | 29.88 | 26.43 |
| AirNet | 26.25 | 23.56 | 26.98 | 14.24 | 27.51 | 30.68 | 28.45 | 23.48 | 24.87 | 30.15 | 25.62 |
| PromptIR | 26.50 | 25.19 | 26.95 | **23.14** | **27.56** | **31.35** | 29.24 | 24.06 | **27.23** | **30.22** | 27.14 |
| IR-SDE | 24.13 | 17.44 | 24.21 | 16.07 | 24.82 | 28.49 | 26.64 | 22.18 | 24.70 | 27.56 | 23.64 |
| Ours | **27.03** | **29.53** | 23.70 | 22.09 | 24.36 | 30.81 | **29.41** | **27.27** | 26.83 | 28.94 | 27.01 |

Table 10: Comparison of our method with other unified image restoration approaches *on SSIM*.

| Model | Blurry | Hazy | JPEG | Low-light | Noisy | Raindrop | Rainy | Shadowed | Snowy | Inpainting | Average |
|---|---|---|---|---|---|---|---|---|---|---|---|
| NAFNet | 0.804 | 0.926 | 0.780 | 0.809 | 0.768 | 0.924 | 0.848 | 0.839 | 0.869 | 0.901 | 0.847 |
| NAFNet+DA-CLIP | **0.816** | **0.948** | **0.798** | 0.823 | **0.777** | 0.928 | 0.869 | **0.849** | 0.880 | 0.914 | **0.861** |
| Restormer | 0.811 | 0.915 | 0.781 | 0.815 | 0.762 | 0.928 | 0.862 | 0.836 | 0.877 | 0.912 | 0.850 |
| AirNet | 0.805 | 0.916 | 0.783 | 0.781 | 0.769 | 0.926 | 0.867 | 0.832 | 0.846 | 0.911 | 0.844 |
| PromptIR | 0.815 | 0.933 | 0.784 | **0.829** | 0.774 | **0.931** | **0.876** | 0.842 | **0.887** | **0.918** | 0.859 |
| IR-SDE | 0.730 | 0.832 | 0.615 | 0.719 | 0.640 | 0.822 | 0.808 | 0.667 | 0.828 | 0.876 | 0.754 |
| Ours | 0.810 | 0.931 | 0.532 | 0.796 | 0.579 | 0.882 | 0.854 | 0.811 | 0.854 | 0.894 | 0.794 |

Table 11: Comparison of our method with other unified image restoration approaches *on LPIPS*.

| Model | Blurry | Hazy | JPEG | Low-light | Noisy | Raindrop | Rainy | Shadowed | Snowy | Inpainting | Average |
|---|---|---|---|---|---|---|---|---|---|---|---|
| NAFNet | 0.284 | 0.043 | 0.303 | 0.158 | **0.216** | 0.082 | 0.180 | 0.138 | 0.096 | 0.085 | 0.159 |
| NAFNet+DA-CLIP | 0.261 | 0.034 | 0.284 | 0.140 | 0.218 | 0.083 | 0.146 | 0.135 | 0.083 | 0.071 | 0.145 |
| Restormer | 0.282 | 0.054 | 0.300 | 0.156 | 0.215 | 0.083 | 0.170 | 0.145 | 0.095 | 0.072 | 0.157 |
| AirNet | 0.279 | 0.063 | 0.302 | 0.321 | 0.264 | 0.095 | 0.163 | 0.145 | 0.112 | 0.071 | 0.182 |
| PromptIR | 0.267 | 0.051 | 0.269 | 0.140 | 0.230 | 0.078 | 0.147 | 0.143 | 0.082 | 0.068 | 0.147 |
| IR-SDE | 0.198 | 0.168 | **0.246** | 0.185 | 0.232 | 0.113 | 0.142 | 0.223 | 0.107 | 0.065 | 0.167 |
| Ours | **0.140** | **0.037** | 0.317 | **0.114** | 0.272 | **0.068** | **0.085** | **0.118** | 0.072 | **0.047** | **0.127** |

Table 12: Comparison of our method with other unified image restoration approaches *on FID*.

| Model | Blurry | Hazy | JPEG | Low-light | Noisy | Raindrop | Rainy | Shadowed | Snowy | Inpainting | Average |
|---|---|---|---|---|---|---|---|---|---|---|---|
| NAFNet | 42.99 | 15.73 | 71.88 | 73.94 | 82.08 | 56.43 | 86.35 | 47.32 | 35.76 | 44.32 | 55.68 |
| NAFNet+DA-CLIP | 36.36 | 11.80 | 68.60 | 71.80 | 79.07 | 43.34 | 66.50 | 37.86 | 29.19 | 34.93 | 47.94 |
| Restormer | 39.08 | 15.34 | 72.68 | 78.22 | 87.14 | 50.97 | 78.16 | 48.33 | 33.45 | 36.96 | 54.03 |
| AirNet | 41.23 | 21.91 | 78.56 | 154.2 | 93.89 | 52.71 | 72.07 | 64.13 | 36.99 | 32.93 | 64.86 |
| PromptIR | 36.5 | 10.85 | 73.02 | 67.15 | 84.51 | 44.48 | 61.88 | 43.24 | 28.29 | 32.69 | 48.26 |
| IR-SDE | 29.79 | 23.16 | 61.85 | 66.42 | 79.38 | 50.22 | 63.07 | 50.71 | 34.63 | 32.61 | 49.18 |
| Ours | **14.13** | **5.66** | **42.05** | **52.23** | **64.71** | **38.91** | **52.78** | **25.48** | **27.26** | **25.73** | **34.89** |

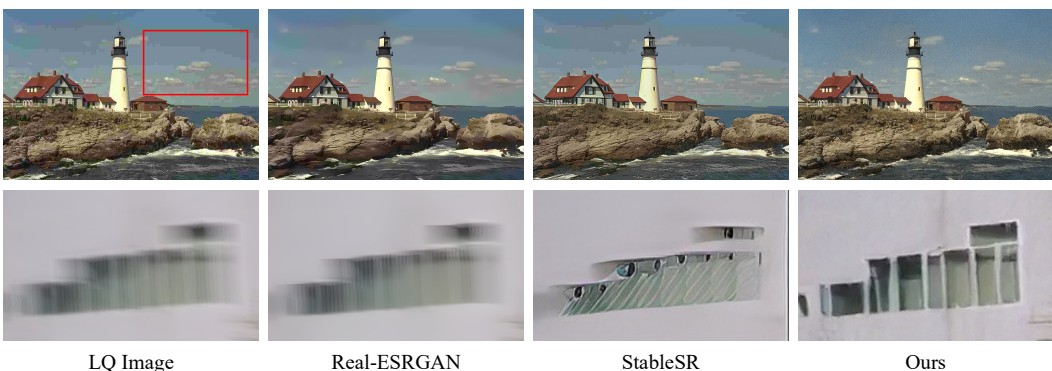

HQ Image    *Blurry* Input    AirNet

*PromptIR*    *IR-SDE*    *Ours*

*Hazy* Input

*JPEG* Input

*Low-light* Input

HQ Image    *Noisy* Input    *PromptIR*    *IR-SDE*    *Ours*

Figure 14: Comparison of our method with other approaches on the *unified* image restoration.

LQ Image    Real-ESRGAN    StableSR    Ours

Figure 15: Comparison of our method with Real-ESRGAN and StableSR.

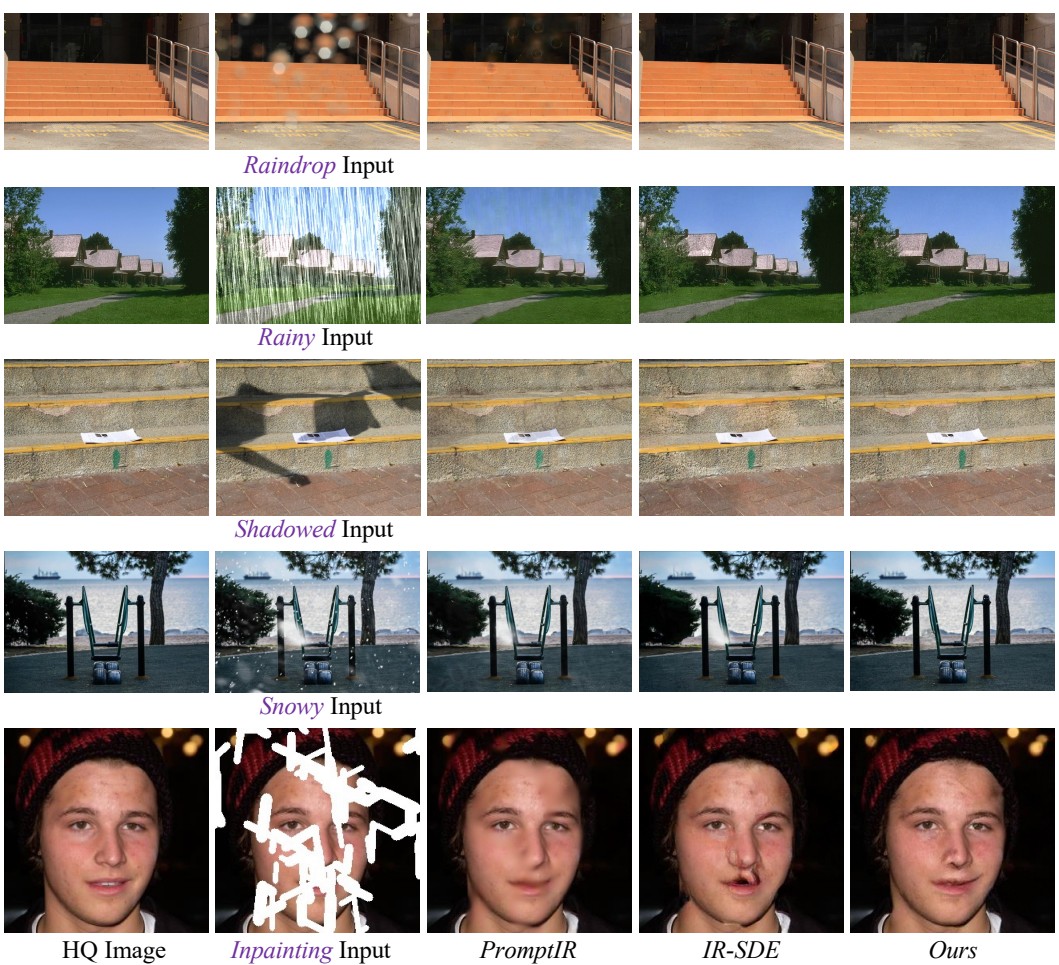

*Raindrop* Input

*Rainy* Input

*Shadowed* Input

*Snowy* Input

| HQ Image | *Inpainting* Input | *PromptIR* | *IR-SDE* | *Ours* |

Figure 16: Comparison of our method with other approaches on the *unified* image restoration.

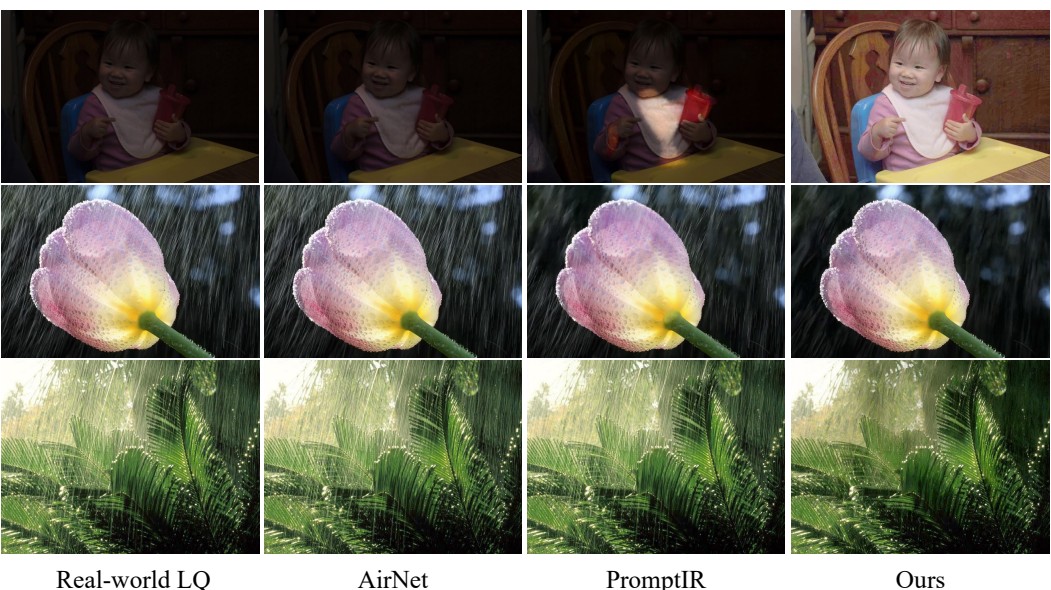

| Real-world LQ | AirNet | PromptIR | Ours |

Figure 17: Comparison of our method with other approaches on *real-world* images.

