# OpenReview forum: "Controlling Vision-Language Models for Multi-Task Image Restoration"
_ICLR.cc/2024/Conference — ICLR 2024 poster_

### Official Review · Reviewer_bHgA · 2023-10-29

**Soundness:** 3 good
**Presentation:** 4 excellent
**Contribution:** 3 good
**Rating:** 6
**Confidence:** 4

**Summary:**

This paper presents a novel method for universal image restoration. The method, named as DA-CLIP, is based on the CLIP and diffusion model (IR-SDE). An additional controller is proposed to help predict high-quality content embedding and degradation type embedding. These two embeddings are then used through cross attention in a diffusion unet to help restore the degraded images. The proposed method is tested on the image denoising, inpainting, deblurring, etc. The experimental results show that the proposed method can achieve better performance than the state-of-the-art methods.

**Strengths:**

1. The paper proposes a novel framework, DA-CLIP, to learn high-quality content and degradation embeddings through contrastive learning.
1. The integration of the degradation and content prompts to universal image restoration appears effective and innovative.
1. Extensive experiments on various tasks are conducted to show the effectiveness of the proposed method.

**Weaknesses:**

1. There are no justifications about what is embedded in the HQ content embedding. It is better to provide some comparisons with the HQ content embedding and the original image embedding.
1. The effectiveness and necessity of the prompt learning module is not well discussed. It is better to provide some ablation studies to show the effectiveness of the prompt learning module compared with naive cross-attention.
1. The performance compared with `NAFNet+DA-CLIP` is not superior.
1. Although the paper discussed the computation complexity in supplementary material, it only provides #params and FLOPS. It is known that the FLOPS is not a good metric for the computation complexity, especially for diffusion models which require multiple iteration steps. The paper should provide the inference time for the proposed method, and better in main paper.
1. Experiments on more complex degradation scenarios, involving multiple concurrent degradation types, would emphasize the model's robustness and versatility.
1. A deeper discussion about the text encoder's role in the performance could lead to a better understanding of the proposed framework.

**Questions:**

#### **1. Why VLM like CLIP is necessary in this paper ?**

After reading the paper, I think that the proposed method does not have much relationship with VLM like CLIP. The DA-CLIP serves as a degradation type classifier and a content classifier. It seems OK to replace CLIP with simple CNNs such as ResNet. Given that the degradation types are quite limited in this paper, a text-encoder like CLIP is easily to be replaced with simple classification. As for the captions, we may also simply replace it with just clean image embeddings to perform contrastive learning.

#### **2. Why diffusion model is used for restoration process ?**

Given the slow inference of diffusion model, it is not clear why diffusion model is used for restoration process. After all, the proposed method has little relationship with diffusion model, especially when the diffusion network is trained from scratch. Compared with `NAFNet+DA-CLIP`, the diffusion based backbone does not have much advantages.

#### **Summary & Conclusion**
In summary, I think that using contrastive learning to learn degradation type embedding and content embedding for universal image restoration is a good idea. And the experiments are also quite comprehensive and effective. However, the proposed integration with VLM and diffusion models are not well justified. And I do not think it is appropriate to claim vision-language models as the main contribution and novelty of this paper.

To conclude, I would like to give borderline to this paper. However, there is no such options in the review form. So I choose to give a marginal accept. I hope the authors can address my concerns in the rebuttal.

---

> ### Author Response · Authors · 2023-11-20
> **Response to Reviewer bHgA (part 1/2)**
>
> Thank you for your insightful review and valuable feedback. We address your questions in turn below:
>
> > Q1: There are no justifications about what is embedded in the HQ content embedding. It is better to provide some comparisons with the HQ content embedding and the original image embedding.
>
> Thank you for this great suggestion. *In Figure 9(c) and 9(d) of the revised manuscript*, we have added the comparison of applying LQ content embeddings (the image embeddings obtained from the original CLIP) to the baseline model on both unified and degradation-specific image restoration tasks. Our DA-CLIP content embeddings clearly outperform the baseline of using original image embeddings. Moreover, the predicted HQ content embedding can also improve MSE-based approaches, please refer to the "General Response" (Table 2 and Table 3) for more details.
>
>
> > Q2: The effectiveness and necessity of the prompt learning module is not well discussed. It is better to provide some ablation studies to show the effectiveness of the prompt learning module compared with naive cross-attention.
>
> Thanks for this nice suggestion. Since the prompt is applied to all blocks of our model, we cannot substitute it with cross-attention which is really computationally inefficient (see *Section 3.2*). Instead, we included a comparison of replacing the prompt module with a simple network *in Figure 9(h) of the revised manuscript*, and the result demonstrates the added benefit of the prompt learning module.
>
>
> > Q3: The performance compared with `NAFNet+DA-CLIP` is not superior.
>
> We need to clarify that the `NAFNet+DA-CLIP` is also our method, which is proposed to prove that DA-CLIP can also improve the performance of vanilla MSE-based restoration approaches. For more details please refer to the "General Response" Table 2 and Table 3. Moreover, as we mentioned in the model evaluation section, this paper mainly focuses on perceptual metrics (which means realistic visual results) and our diffusion-based method performs the best in terms of LPIPS and FID. As further illustrated in the face inpainting case *in Figure 8*, our DA-CLIP generates correct details for the left eye region while other MSE-based methods can only produce blurry and smooth contents.
>
>
> > Q4: It is known that the FLOPS is not a good metric for the computation complexity, especially for diffusion models which require multiple iteration steps. The paper should provide the inference time for the proposed method, and better in the main paper.
>
> Thanks for pointing this out. It is true that diffusion-based methods require more inference time for multiple iterations. We have now added the runtime comparison in the below table. Note that MAXIM is implemented with the JAX GPU version, thus its comparison may not be fair. We then add the PromptIR and NAFNet for further reference. For the diffusion baseline IR-SDE, our DA-CLIP only adds 0.06s over 100 iterations (the prompt module and cross-attention also add 0.3s). For the direct MSE-based baseline NAFNet, there is no obvious increase in inference time.
>
> **Table 4**. Comparison of the number of parameters and model computational efficiency. The flops and inference time are computed on the inpainting dataset.
>
> |  Method  |  MIXIM  |  PromptIR  |  NAFNet  |  NAFNet + DA-CLIP  |  IR-SDE  |  IR-SDE + DA-CLIP  |
> |  ----  | ----  |  ----  | ----  |  ----  |  ----  | ----  |
> | \#Parameters | 14.1M | 33M | 67.8M | 86.1M + 125.2M | 36.2M | 48.9M + 125.2M |
> | FLOPs | 216G | 158G | 63G | 65G + 0.5G | 117G | 118G + 0.5G |
> | Inference time | 2.9s | 0.1s | 0.08s | 0.08s | 4.2s | 4.53s + 0.06s |
>
> ---

---

> > ### Author Response · Authors · 2023-11-20
> > **Response to Reviewer bHgA (part 2/2)**
> >
> > > Q5: Experiments on more complex degradation scenarios, involving multiple concurrent degradation types, would emphasize the model's robustness and versatility.
> >
> > Since the mixed degradation dataset only contains a single degradation type label for each image, it is difficult for our method (and all other unified models) to restore multiple degradations of the same scene. Thank you for this comment, we would be very interested in creating models that can recover multiple concurrent degradations in future work. Moreover, as we mentioned in the "General Response" Table 1, DA-CLIP outperforms other unified methods on the light rain dataset (an out-of-distribution degradation), which also illustrates the robustness and versatility of our model.
> >
> > > Q6: A deeper discussion about the text encoder's role in the performance could lead to a better understanding of the proposed framework.
> >
> > The text encoder is always fixed and only used for training DA-CLIP to predict degradation embeddings and HQ content embeddings. However, we can directly integrate the text encoder's embedding into the downstream model as a text-guided restoration process. The experiments are shown *in Figure 9(f) and Figure 9(g) in the revised manuscript*, in which we find that using ground truth text embeddings from the text encoder leads to a similar performance to that of using image embeddings from our DA-CLIP. However, text embedding requires access to the ground truth degradation label and clean caption for each image. Since we only have LQ images in testing, we choose to use image embeddings for all experiments.
> >
> > > Q7: Why VLM like CLIP is necessary in this paper?
> >
> > The main motivation for leveraging CLIP in this work is to provide meaningful semantic instructions to improve the restoration process. Previous works like AirNet [1] have tried to use a CNN encoder with contrastive learning as the degradation classifier, but its performance is much worse than our DA-CLIP, as illustrated *in Table 3 of our paper*. Particularly, CLIP is pretrained on billions of web-scale text-image pairs which can provide more accurate visual embeddings and stronger generalization ability than vanilla-trained CNNs or vision Transformers. As we mentioned in the "General Response" Table 1, our DA-CLIP performs much better than all other unified models on the unseen light rain dataset.
> >
> > > Q8: Why diffusion model is used for the restoration process?
> >
> > As we clarified in Q3, we focus on the diffusion-based model due to its strong ability to generate realistic textures and details (which means better visual results, as for the face inpainting example *in Figure 8*). Moreover, we clarify that our DA-CLIP also works well for vanilla MSE-based approaches, such as the `NAFNet+DA-CLIP`. Please refer to the "General Response" Table 2 and Table 3 for more details.
> >
> > ---
> >
> > *[1] All-in-one Image Restoration for Unknown Degradations Using Adaptive Discriminative Filters for Specific Degradations. CVPR 2023.*

---

> ### Comment · Reviewer_bHgA · 2023-11-21
>
> The authors are appreciated for providing clarifications and additional experiments regarding my concerns. They have addressed some of the questions, such as Q1. However, the responses to other critical inquiries remain unsatisfactory.
>
> - **Generalization ability.**  As pointed out by other reviewers, the proposed method and provided codes does not show superior performance with complex degradations, which might be the true home field of diffusion models. Although in general response, the author provides experiment of light rain as OOD experiment. It is still a quite simple case compared with complex degradations used in related works such as StableSR, DiffBIR etc.
>
> - **Motivation and Advantages of Utilizing VLM.** The author responses that ``CLIP provides more accurate visual embeddings and stronger generalization ability``. However, it is worth noting that other pretrained backbones, like DINO, may yield more robust visual embeddings. This explanation does not entirely justify the selection of CLIP in this context. Additionally, the learning pipeline of DA-CLIP may not necessitate linguistic components due to the simplicity of the degradation types involved. For instance, a similar classifier, such as AirNet, could potentially perform just as effectively. It is my belief that the advantages and potential versatility of language components remain underexplored in this study. It would be more intriguing if the method could generalize to more complex degradations, such as "snowy and rainy," as this could be the key to justifying the inclusion of a text encoder: the ability to specify various degradation types using natural language.
>
> - **The usage of diffusion model.** While the diffusion model does lead to improved perceptual quality (measured by lpips and fid), the enhancement is marginal and not visually obvious. Even the disparities in results presented in Figure 8 are relatively minor, and there is no presentation of results for NAFNet+DACLIP. Considering that the diffusion model substantially increases computational cost and runtime compared to NAFNet, one must question why diffusion model is necessary here. While I acknowledge that NAFNet in conjunction with DACLIP is also your results, if the diffusion model does not exhibit superiority, it might be better to consider removing it from the paper.
>
> In conclusion, the provided responses have not adequately addressed my primary concerns, and I maintain my original rating preference.

---

> ### Author Response · Authors · 2023-11-22
> **Official Response by Authors**
>
> Dear Reviewer bHgA,
>
> Thank you for your reply. We are pleased to know that our new experiments have addressed some of your questions. In the following we further address your remaining concerns:
>
> > Q(a): Generalization ability.
>
> We would like to note that the blind approaches like StableSR and DiffBIR can recover images with mixed blur, compression, and noise because they are specifically trained on these mixed degradations, see the argument in [1, Fig.2]. It is undoubtedly interesting to create models to handle concurrent degradations like “rain + shadow”, “snow + low-light”, “haze + raindrop”, etc. However, mixing these degradations is hard since most of them are captured from specific scenes and camera devices, and manually synthesizing these degradations requires knowing the corresponding domain-specific knowledge (i.e., the true degradation function). We agree that our generalization ability thus is not perfect (but is still better than other all-in-one methods such as AirNet and PromptIR). Considering this, we have revised the description of our method from “universal” to “multi-task”, and we have also emphasized the limitations in the revised manuscript. The light rain experiment evidences that our method is robust to different degradation levels. In addition, we provided additional real-world restoration examples in Figure 16 for reference, which also illustrates that our method has better generalization ability than peer all-in-one approaches.
>
> > Q(b): Motivation and Advantages of Utilizing VLM.
>
> Incorporating pretrained foundation models (e.g., CLIP, DINO) for low-level vision tasks such as image restoration is still an open problem. In this paper, we focus on the VLM for accurate semantic instructions which are also used in other text-guided image generation models like Stable Diffusion, ControlNet, Imagen, InstructPix2Pix, etc.  However, our core component, the image controller, is applicable to any other backbones (e.g., DINO) that have a ViT-based image encoder, as shown in Figure 3(a). We appreciate the reviewer’s feedback and suggestions for more robust backbones and we will definitely test them in future work.
>
> Moreover, we also appreciate the feedback on linguistic components which is really valuable and inspires our future work! As we explained in Q(a), our method is not able to properly process mixed degradations in the current work. We thus agree with the reviewer that the linguistic components are under-explored and we have then added it to our limitations. However, we would like to clarify that our method not only provides the ability of degradation classification but also predicts the clean content embedding which can further improve the performance. While other simple classifiers can hardly do this since there are no ground truth HQ embeddings in the dataset.
>
> > Q(c): The usage of diffusion model.
>
> Image restoration methods are usually sensitive to image details and textures. It has been shown that MSE-based models tend to produce blurry results to get high peak signal-to-noise ratios (PSNR), see the arguments in [2, pp.1] and [3, pp.1]. Previous works [2,3] address this issue by incorporating adversarial networks, which are however notoriously unstable and inaccurate [4, pp.1]. Thus, most recent works tend to utilize diffusion models to generate realistic outputs [4,5,6].
>
> The visual results for `NAFNet+DACLIP` are very similar to other MSE-based methods like AirNet and PromptIR. As suggested, we will add more visual comparisons to the Appendix in our revision. Moreover, we would like to clarify that our improvement in perceptual scores is significant compared to other MSE-methods. In particular, our method surpasses the MAXIM and PromptIR by ~47 and ~14 in terms of FID in Table 2(a) and Table 3, respectively (from 58.7 to 11.7 in Table 2(a), from 48.2 to 34.8 in Table 3). Apart from the inpainting case in Figure 8 (left eye region), Figure 5 (deraining) and Figure 15 (inpainting) also illustrate that MSE-based approaches tend to produce blurry and over-smooth visual results (in degraded areas) compared to our method.
>
> Moreover, choosing a specific restoration model (diffusion- or MSE-based) is not our key component. In this paper, we would like to include both `IR-SDE+DA-CLIP` and `NAFNet+DA-CLIP` to demonstrate that our DA-CLIP can be integrated with various restoration models, to demonstrate the general applicability in that sense.
>
> ---
>
> *[1] DiffBIR: Towards Blind Image Restoration with Generative Diffusion Prior. ArXiv 2023.*
>
> *[2] Photo-Realistic Single Image Super-Resolution Using a Generative Adversarial Network. CVPR 2017.*
>
> *[3] Image-to-image translation with conditional adversarial networks. CVPR 2017.*
>
> *[4] Image super-resolution via iterative refinement. TPAMI 2022.*
>
> *[5] Denoising Diffusion Restoration Models. NeurIPS 2022.*
>
> *[6] Image restoration with mean-reverting stochastic differential equations. ICML 2023.*

---

> > ### Author Response · Authors · 2023-11-23
> > **A Kind Reminder - Response period coming to an end**
> >
> > Dear Reviewer,
> >
> > This is a friendly reminder that the response period is coming to an end. We hope that our rebuttal and the revised manuscript have addressed most of your concerns.
> >
> > Thank you again for your valuable feedback and comments.
> >
> > Best regards,
> >
> > Authors

---

### Official Review · Reviewer_3ZRs · 2023-10-30

**Soundness:** 3 good
**Presentation:** 3 good
**Contribution:** 3 good
**Rating:** 6
**Confidence:** 5

**Summary:**

This paper presents a degradation-aware vision language model (DA-CLIP) to generate high-quality image representation and distinct the degradation types of low-quality inputs for all-in-one image restoration. The DA-CLIP model can be integrated into different image restoration networks to improve the performance. Extensive experiments are conducted to demonstrate the effectiveness of the proposed method.

**Strengths:**

1. The idea of constructing a vision-language model to restore clean semantic image representation and distinct degradation types of low-quality images is interesting.
2. The method of using clean image representative and degradation prompt to instruct restoration networks for better performance is sound.
3. The results look good, and the experimental analysis demonstrate the effectiveness of the DA-CLIP on all-in-one image restoration.
4. The writing is well, and the paper is easy to read.

**Weaknesses:**

1. It is questionable that the caption embedding can provide a high-quality image representation supervision for the content embedding. $e_c^T$ can indeed provide a semantic supervision, but there is no guarantee that it is a clean image representation. Therefore, I think the claim that the image encoder with the controller outputs high-quality content features is not rigorous. It seems that $e_c^I$ mainly serves to provide semantic instruction for the restoration network, especially for diffusion-based models.
2. The used experimental setup is too simple to demonstrate the superiority of this complex method. It is not difficult for a unified network, e.g., a vanilla version Restormer, to deal with the all-in-one image restoration setting with a specific degradation level for each degradation type.
3. This paper do not provide the experiment about the generalization ability of the proposed method. I do some tests using the code provided by the authors, and the results show that the model cannot deal with out-of-distribution degradations as well as OOD degradation levels well. It is not surprising as the used degradation model is too simple. This also reflects that $e_c^I$ is not always a high-quality image representation.
4. This method may be difficult to handle tasks with different degradation levels, because it is difficult to describe the specific degradation level in texts. Since the authors do not provide relevant experiments, the potential of this method to handle multiple degradation levels is still questionable. As far as the current results are concerned, the approach is not practical enough.

**Questions:**

1. Do both $e_c^I$ and $e_d^I$ have an important impact on the restoration performance? Intuitively, it is reasonable for diffusion-based models as $e_c^I$ controls the content and $e_d^I$ indicates the degradation. However, it seems not reasonable for mse-based models to use $e_c^I$ in the restoration process.
2. What performance can be achieved by directly providing the semantic text prompt and the degradation type prompt to train the all-in-one image restoration diffusion model?
3. What performance can be achieved by directly training a vanilla Restormer under the same all-in-one setting?

---

> ### Author Response · Authors · 2023-11-20
> **Response to Reviewer 3ZRs (part 1/2)**
>
> Thank you for your comprehensive and insightful review. We address your questions in turn below:
>
> > Q1: It is questionable that the caption embedding can provide a high-quality image representation supervision for the content embedding. [...] It seems that $e_c^I$ mainly serves to provide semantic instruction for the restoration network, especially for diffusion-based models.
>
> A key motivation for leveraging the pretrained VLM for image restoration is that VLMs like CLIP can accurately align image representations with corresponding caption embeddings. Note that the caption generated from an LQ image would be different from that of an HQ image. Such as the snow example shown *in Figure 4*, the caption from HQ image precisely describes the main content of the scene, which supervises our model to learn a snow-free HQ image representation from the LQ image. In addition, since we fixed the text encoder in training, matching the heavy snow scene to a snow-free caption embedding guarantees that the learned content embedding contains no snow information, which is called "clean" and "high-quality" in the context of image desnowing.
>
> We have further illustrated this by applying two different image content embeddings (HQ embedding from DA-CLIP and LQ embedding from the original CLIP) to the baseline *in Figures 9(c) and 9(d) of the revised submission*. The results show that our DA-CLIP content embedding can provide more useful supervision to the learning process on both unified and degradation-specific restoration.
>
> Moreover, the learned content embedding $e_c^I$ can also benefit MSE-based restoration approaches. As we illustrated in the "General Response" Table 2 and Table 3, both NAFNet and PromptIR can be further improved by adding our DA-CLIP content embeddings.
>
> > Q2: The used experimental setup is too simple to demonstrate the superiority of this complex method. It is not difficult for a unified network, e.g., a vanilla version Restormer, to deal with the all-in-one image restoration setting with a specific degradation level for each degradation type.
>
> Although we can directly train a vanilla network across multiple degradations, its performance is inferior to approaches that learn discriminative degradation context explicitly [1,2]. As illustrated in the "General Response" Table 2, combining the vanilla model NAFNet with our DA-CLIP significantly outperforms the original NAFNet on the unified restoration task. In addition, as nicely suggested, we have added the Restormer experiment *in Table 3 of the revised manuscript* (also in the "General Response" Table 2). It is worth noting that PromptIR is built upon Restormer with additional degradation prompt modules, which largely improves the unified restoration results.
>
> ---
>
> *[1] All-in-one Image Restoration for Unknown Degradations Using Adaptive Discriminative Filters for Specific Degradations. CVPR 2023.*
>
> *[2] PromptIR: Prompting for All-in-One Blind Image Restoration. NeurIPS 2023.*

---

> > ### Author Response · Authors · 2023-11-20
> > **Response to Reviewer 3ZRs (part 2/2)**
> >
> > > Q3: This paper does not provide the experiment about the generalization ability of the proposed method. [...] the model cannot deal with out-of-distribution degradations as well as OOD degradation levels well.
> >
> > Due to the data source limitation, almost all unified models struggle to deal with OOD degradations. In the "General Response" Table 1, we provided an experiment of testing all models on the Rain100L dataset which contains 100 light rain images. Although our method is only trained on heavy rain images (Rain100H), it still performs well and significantly surpasses all other unified approaches. Moreover, we have added visual results on real-world images *in our revised submission (Figure 16)*, which also indicate that our method has a better generalization ability compared to other unified methods. In our future work, we could potentially add some data augmentation steps (e.g., resize, random noise, and blur) in training to make our model more practical, similar to other blind super-resolution approaches such as Real-ESRGAN. We would be also very interested in creating models that perform well on OOD degradation types and levels.
> >
> > > Q4: This method may be difficult to handle tasks with different degradation levels, because it is difficult to describe the specific degradation level in texts.
> >
> > It is true that DA-CLIP cannot describe the degradation levels in texts. However, benefiting from the predicted HQ content embedding, our method is still able to handle different degradation levels. For example, both the hazy and snowy datasets contain mixed images across multiple degradation levels. Moreover, in the "General Response", Table 1 also illustrates that our DA-CLIP generalizes better to an unseen degradation level compared to other unified models.
> >
> > > Q5: Do both $e_c^I$ and $e_d^I$ have an important impact on the restoration performance? [...] However, it seems not reasonable for mse-based models to use $e_c^I$ in the restoration process.
> >
> > In our experiments, both $e_c^I$ and $e_d^I$ are important and can improve the performance of unified models as illustrated *in Figures 9(a), 9(c) and 9(d) in the revised manuscript*. Moreover, the two embeddings can be applied also to MSE-based restoration models, as illustrated by the ablation experiments in Table 2 and Table 3 in the "General Response". The results show that adding DA-CLIP content embedding $e_c^I$ to NAFNet and PromprIR improves their performance, illustrating the importance of DA-CLIP content embedding on MSE-based approaches.
> >
> > > Q6: What performance can be achieved by directly providing the semantic text prompt and the degradation type prompt to train the all-in-one image restoration diffusion model?
> >
> > The model with ground truth degradation text embeddings has a similar performance to the predicted degradation embeddings, as illustrated *in Figure 9(e) of the revised manuscript*. Moreover, we further added a comparison of providing both ground truth (GT) degradation type and semantic caption text embeddings to the baseline diffusion model *in Figure 9(f) and 9(g)*. The model with both GT degradation and caption text embeddings outperforms the baseline that only has degradation text embeddings (*Figure 9(f)*). However, the method with GT text embeddings performs slightly worse than our model which uses predicted image embeddings (*Figure 9(g)*).
> >
> > > Q7: What performance can be achieved by directly training a vanilla Restormer under the same all-in-one setting?
> >
> > As shown in the "General Response" Table 2, Restormer performs slightly better than NAFNet but is inferior to PromptIR. In fact, PromptIR is built on top of Restormer, simply adding degradation prompt modules. In addition, Table 2 also shows that adding DA-CLIP content embedding can further improve the performance across all metrics.

---

> > > ### Author Response · Authors · 2023-11-23
> > > **A Kind Reminder - Response period coming to an end**
> > >
> > > Dear Reviewer,
> > >
> > > This is a friendly reminder that the response period is coming to an end. We hope that our rebuttal and the revised manuscript have addressed most of your concerns and the reviewer could reassess our work and consider updating their score.
> > >
> > > Furthermore, we are happy to address any outstanding questions and thank you again for your valuable feedback and positive review.
> > >
> > > Best regards,
> > >
> > > Authors

---

### Official Review · Reviewer_hnNQ · 2023-11-04

**Soundness:** 2 fair
**Presentation:** 3 good
**Contribution:** 2 fair
**Rating:** 3
**Confidence:** 5

**Summary:**

This paper proposes a degradation-aware CLIP model. It is aligned with language through dual aspects of image content and degradation during training. This DA-CLIP can extract information not only about the image content but also about image degradation. The authors also combined DA-CLIP with image restoration, proposing what they call a "universal" image restoration method. This method is based on Diffusion-Based restoration techniques, but the DA-CLIP's results are used as controlled prompts for input. The authors have showcased many results.

**Strengths:**

I hold a positive view on the idea of incorporating degradation information into CLIP.

**Weaknesses:**

My main concern with this paper is its task setting. First, "Universal Image Restoration" is a term that is not so easily justified. This paper simply brings together ten different image restoration tasks, which is closer to "multi-task" than the so-called "universal". For a large model, mixing these ten tasks in such a separate manner for training, the model would internally categorize the problems before handling them in a single-task manner [R1]. This would not endow the model with sufficient generalization capabilities. For instance, an image with both rain streaks and subsequent compression artifacts cannot be accurately restored. This is not "universal". Moreover, this paper seems to ignore a host of more "universal" solutions, such as Real ESRGAN, BSRGAN, StableSR, DiffBIR, etc. Merging degradations to achieve better generalization is a new direction (which is not so new anymore). But this paper barely discusses whether these methods are "universal" or not.

Secondly, DA-CLIP's ability to predict degradation is not particularly special, considering the task setting only involves ten types of degradation; it can be said that almost any image restoration model trained on these degradations or any model that understands or classifies them would have this ability [R2]. I can only say that introducing degradation into CLIP is a very promising direction and could be very useful. However, the approach taken in this paper fails to reflect any significance in doing so. Due to the inherent issues with the task setting of the paper, the experimental part also fails to demonstrate the corresponding contributions.

[R1] Finding Discriminative Filters for Specific Degradations in Blind Super-Resolution
[R2] Discovering “Semantics” in Super-Resolution Networks

**Questions:**

See Weakness

---

> ### Author Response · Authors · 2023-11-20
> **Response to Reviewer hnNQ (part 1/2)**
>
> Thank you for providing valuable feedback for our paper. We address your concerns in turn below. We sincerely hope that this response addresses all the concerns and that the reviewer would consider re-evaluating our work. We look forward to and appreciate your further feedback, we would be happy to provide further clarifications if needed.
>
> > Q1: My main concern with this paper is its task setting. First, "Universal Image Restoration" is a term that is not so easily justified. This paper simply brings together ten different image restoration tasks, which is closer to "multi-task" than the so-called "universal". [...] For instance, an image with both rain streaks and subsequent compression artifacts cannot be accurately restored. This is not "universal".
>
> Note that the "universal" means our method is a general framework in which the proposed DA-CLIP can be applied to improve both degradation-specific and unified image restoration models, as illustrated *in Table 2 and Table 3 of the manuscript*. We have rewritten the introduction to clarify this in the revised manuscript. It is also worth noting that all existing unified models have been evaluated only on three or four degradation types, while our method handles ten restoration tasks with various input sizes and complicated modalities (as shown *in Figure 1*). In this situation, learning discriminative degradation context explicitly has been proven a vital step for unified image restoration [1,2,3].
>
> In addition, please note that none of the existing unified models trained on our dataset can restore images with multiple specific degradations (such as a scene that mixes shadows and raindrops, like the Raindrop example *in Figure 8*), since we have only one degradation type label in each training image pair. That is, our DA-CLIP shares this limitation with all other existing unified restoration models. However, it would indeed be possible to add some simple degradations like noise and blur to synthesize multi-degradation image pairs in training, potentially mitigating this issue. Thank you for your feedback, we would be very interested in creating more practical models that are able to fully restore images with mixed degradation types in our future work.
>
> > Q2: Moreover, this paper seems to ignore a host of more "universal" solutions, such as Real ESRGAN, BSRGAN, StableSR, DiffBIR, etc. Merging degradations to achieve better generalization is a new direction (which is not so new anymore). But this paper barely discusses whether these methods are "universal" or not.
>
> We need to clarify that approaches like Real-ESRGAN, BSRGAN, StableSR, and DiffBIR are not "universal" since most of them mainly focus on the image super-resolution problem. They are indeed practical for some blind degradation conditions such as noise and blur, but can never recover images from more complicated scenarios such as shadow removal, deraining, inpainting, etc. Moreover, the goal of unified image restoration is not only to provide generalization ability but also to avoid repeated training of individual networks for specific degradation types [3]. As nicely suggested, in our revised manuscript, we have added a discussion on these blind image restoration methods in the related work section.
>
> ---
>
> *[1] Learning Weather-General and Weather-Specific Features for Image Restoration Under Multiple Adverse Weather Conditions. CVPR 2023.*
>
> *[2] All-in-one Image Restoration for Unknown Degradations Using Adaptive Discriminative Filters for Specific Degradations. CVPR 2023.*
>
> *[3] PromptIR: Prompting for All-in-One Blind Image Restoration. NeurIPS 2023.*

---

> > ### Author Response · Authors · 2023-11-20
> > **Response to Reviewer hnNQ (part 2/2)**
> >
> > > Q3: DA-CLIP's ability to predict degradation is not particularly special, considering the task setting only involves ten types of degradation; it can be said that almost any image restoration model trained on these degradations or any model that understands or classifies them would have this ability.
> >
> > As we mentioned in the first question, learning discriminative degradation context explicitly has been proven a vital step for unified image restoration. However, we want to clarify that simply classifying different degradation types is not the key contribution of our method. The goal of DA-CLIP is to **1)** extract accurate discriminative degradation contexts for unified image restoration and **2)** provide meaningful semantic instructions to improve the general restoration process. As we illustrated *in Table 2 in the paper*, DA-CLIP improves the baseline IR-SDE on degradation-specific tasks without classifying degradation types. Moreover, DA-CLIP can improve the generalization ability of the unified model ("General Response Table 1" and *Figure 16 in the revised submission*). We have also provided a comprehensive analysis on two types of embeddings from DA-CLIP *in Figure 9 and Figure 11*. For the motivation of using CLIP please refer to the "General Response" for more details.
> >
> > > Q4: I can only say that introducing degradation into CLIP is a very promising direction and could be very useful. However, the approach taken in this paper fails to reflect any significance in doing so.
> >
> > Thanks for your positive feedback on the underlying idea of DA-CLIP. To the best of our knowledge, this is also the first work combining VLMs with general image restoration methods. Based on the high-accuracy degradation prediction and the ability to output clean content embeddings, DA-CLIP improves the baseline models (IR-SDE and NAFNet) on all degradation-specific tasks (*Table 2, Table 7, and Figure 5*) and on the unified image restoration task (*Table 3, Figure 6, Figure 7, and Figure 8*) across all metrics, demonstrating the effectiveness of our method. In addition, we provided comprehensive ablation experiments on two types of DA-CLIP embeddings: HQ content embedding and degradation embedding in the discussion section (*Figure 9, Figure 11, and Table 6*). We believe that all these results and analyses are sufficient to demonstrate the significance of the proposed DA-CLIP.

---

> > > ### Comment · Reviewer_hnNQ · 2023-11-21
> > > **Response to author response (2/2)**
> > >
> > > Being the "first work" is not a reason why this paper should be accepted. Being a "reasonable" work is. The authors should carefully review developments in the field of image restoration and identify the issues of greatest concern to this field. And be objective and humble about the methods they propose.

---

> > ### Comment · Reviewer_hnNQ · 2023-11-21
> > **Response to author response (1/2)**
> >
> > The main reason I object to this paper is this definition of "universal". Almost all image-to-image deep learning method frameworks can be considered “degradation-specific” and “unified image restoration”. The deep learning method framework itself is a relatively general method framework. This does not prove that the "universal" proposed in the paper is a sufficiently accurate and modest term. The authors argue that existing methods cannot handle degradation such as rain, snow, and fog. First of all, the method proposed by the author cannot achieve good results on mixed degradation and complex degradation, too. Secondly, the mixed training method can also deal with the degradation in the training set if rain, snow and fog are added. Why these methods do not consider rain, fog, and snow is because there is no good method that can give the model good generalization ability in these degradations. This is true for the method proposed in this paper. There is almost no essential difference between them and the method proposed by the author. However, previous methods, due to training on multiple mixture degradations, have acceptable generalization capabilities within the range that these degradations may cover. However, the method proposed in this paper fails to work on quite a few mixture degradations. Even so, these methods only modestly claim to be blind or multitasking. I really don't see why the method in this paper can be called "universal".
> >
> > I can think of two ways this paper could be justified. First, the method is simply described as multi-task image restoration. The focus is on describing the impact of task instructions on recovery results. Second, focus on the description of DA-CLIP. Describe how this method can be used for "multi-tasking" image restoration and some other applications. Either way, there is no support for the word "universal" being proposed.

---

> ### Author Response · Authors · 2023-11-21
> **Response to Reviewer hnNQ**
>
> Dear Reviewer hnHQ,
>
> Thank you very much for your reply and we appreciate your comments on the definition of “universal” and suggestions for the justification.
>
> 1. We were not trying to oversell our method by using the word "universal". We were seeking a suitable term to inform that our method can benefit both degradation-specific and unified (all-in-one) image restorations. We have also emphasized the limitations (e.g., to deal with multiple degradations at the same time) in the resubmission.  We will change the word “universal” in the manuscript to a neutral one, such as multi-task, as per your request, to avoid abusing the term.
>
> 2. We have also reclarified the difference to the suggested related works (e.g., Real-ESRGAN, BSRGAN, StableSR, DiffBIR). Specifically, these works do not make use of the semantic embeddings from the vision-language model, while on the other hand, the gist of our paper is to incorporate such semantic embeddings, and we have shown that it indeed improves the image restoration performance.
>
> 3. We agree with the reviewer that vanilla deep learning frameworks can be directly trained for unified image restoration. However, they are not specifically designed for multi-task learning. Hence, training these methods by directly mixing all degradations ignores learning accurate discriminative degradation contexts, see the arguments in [1, pp.3] and [2, pp.2]. Therefore, the purpose of DA-CLIP is to incorporate additional semantic instructions (degradation and HQ content embeddings) to steer vanilla networks for better performance (such as `IR-SDE+DA-CLIP` and `NAFNet+DA-CLIP`).
>
> 4. We definitely agree that "first work" is not a reason for acceptance, and for that we did not list it as a contribution either. As acknowledged by other reviewers, our proposed DA-CLIP informs the community an interesting and promising direction of combining pretrained vision-language models with image restoration frameworks. Our experiments and analysis have also empirically demonstrated the effectiveness of our method.
>
> Best,
> Authors
>
> ---
>
> *[1] All-in-one Image Restoration for Unknown Degradations Using Adaptive Discriminative Filters for Specific Degradations. CVPR 2023.*
>
> *[2] PromptIR: Prompting for All-in-One Blind Image Restoration. NeurIPS 2023.*

---

> > ### Author Response · Authors · 2023-11-23
> > **Response to Reviewer hnNQ**
> >
> > Dear Reviewer,
> >
> > As we have addressed all your questions and also updated the manuscript as per your suggestions, we hope that the reviewer can reassess our work and consider updating their score.
> >
> > Furthermore, we are happy to address any outstanding questions and thank you again for your valuable review.
> >
> > Best regards,
> >
> > Authors

---

### Official Review · Reviewer_j7ME · 2023-11-04

**Soundness:** 3 good
**Presentation:** 3 good
**Contribution:** 3 good
**Rating:** 6
**Confidence:** 5

**Summary:**

The paper proposes a framework called degradation-aware CLIP (DA-CLIP) that combines large-scale pretrained vision-language models with image restoration networks. The authors address the issue of feature mismatching between corrupted inputs and clean captions in existing vision-language models (VLMs) by an Image Controller that adapts the VLM's image encoder to output high-quality content embeddings aligned with clean captions. The controller also predicts a degradation embedding to match the real degradation types. The paper presents the construction of a mixed degradation dataset for training DA-CLIP and demonstrates its effectiveness in both degradation-specific and unified image restoration tasks. The results show highly competitive performance across ten different degradation types.

**Strengths:**

- This paper proposes a novel framework, DA-CLIP, which combines large-scale pretrained vision-language models with image restoration networks.
- This paper introduces an Image Controller that addresses the feature mismatching issue between corrupted inputs and clean captions in existing vision-language models. In addition, they introduce a prompt learning module to better utilize the degradation context for unified image restoration.
- It demonstrates that DA-CLIP in both degradation-specific and unified image restoration tasks, achieving highly competitive performance across all ten degradation types.

**Weaknesses:**

- In Figure 1, DA-CLIP achieves surprisingly high accuracy in ten degradation types. How are these experiments set up? In contrast, CLIP performs poorly in many types. What prompts do the authors use for classifying degradations in CLIP?
- In Figure 6, PromptIR is comparable or even better than the proposed DA-CLIP in most tasks on fidelity metrics.
- In Table 2(c), the PSNR of DA-CLIP highly deviates from that of MAXIM. In addition, the results on task-specific restoration do not show a clear benefit of using a universal model for all tasks. It is believed that the merit of universal models is that different tasks can benefit each other, or at least be helpful in generalization to new domains.

**Questions:**

See weaknesses

---

> ### Author Response · Authors · 2023-11-20
> **Response to Reviewer j7ME**
>
> Thank you for your review and comments, which accurately summarized our paper and our intended contributions. We provide our point-to-point response below:
>
> > Q1: In Figure 1, DA-CLIP achieves surprisingly high accuracy in ten degradation types. How are these experiments set up? In contrast, CLIP performs poorly in many types. What prompts do the authors use for classifying degradations in CLIP?
>
> In our experiments, we use the ten degradation types as label texts, and the prompt is set to "a *[degradation type]* photo". Since we freeze the text encoder in training, the same text embedding would be used for both DA-CLIP and the original CLIP to match low-quality images. We provide additional experiments of directly retraining or finetuning CLIP on our dataset *in Table 4 of the paper*. Our DA-CLIP achieves a similar accuracy with CLIP + finetune, while it can also output clean content embedding to further improve downstream image restoration performance.
>
> > Q2: In Figure 6, PromptIR is comparable or even better than the proposed DA-CLIP in most tasks on fidelity metrics.
>
> Note that PromptIR is a purely MSE-based approach that achieves high fidelity metric scores (PSNR and SSIM) but will produce perceptually unsatisfying results[1]. Our method, similar to other diffusion approaches[2], mainly focuses on generating realistic images and achieves the best perceptual performance (LPIPS and FID) *in Figure 6 and Table 3 of the paper*. As can be seen in the inpainting example in *Figure 8*, the face generated by PromptIR loses all details on the left eye region, while our method predicts more details and produces relatively good visual results. In addition, simply adding the HQ content embedding of DA-CLIP can further improve PromptIR across all metrics (please refer to the "General Response" Table 2 for more details).
>
> > Q3: In Table 2(c), the PSNR of DA-CLIP highly deviates from that of MAXIM.
>
> Similar to PromptIR, MAXIM is also an MSE-based method and is designed for degradation-specific image restoration. Although it surpasses DA-CLIP by 2dB in terms of PSNR on the GoPro dataset, our method still obtains the best perceptual scores (0.058/6.15 in LPIPS/FID compared to 0.089/11.57) and produce visually better results. In the "General Response" Table 3, we have further added an ablation experiment of applying DA-CLIP to NAFNet on two single-degradation tasks: image deraining and low-light enhancement. The results show that DA-CLIP clearly improves the performance of NAFNet, illustrating the effectiveness of our method on degradation-specific restoration.
>
> > Q4: In addition, the results on task-specific restoration do not show a clear benefit of using a universal model for all tasks. It is believed that the merit of universal models is that different tasks can benefit each other, or at least be helpful in generalization to new domains.
>
> We need to clarify that separately training models for each task often gives better restoration performance than the unified setting, as each model then only needs to focus on one degradation throughout the training. This phenomena has been discussed in more detail in recent previous work [3, 4]. The main motivation for the unified model is **1)** to avoid repeated training of the same model for individual tasks and **2)** to avoid the need to know the specific input degradation type in order to apply the relevant model (which is usually required by task-specific methods[3, 4, 5]). Thank you for this valuable feedback, we have now clarified this motivation in the revised introduction section.
>
> ---
>
> *[1] Photo-realistic single image super-resolution using a generative adversarial network. CVPR 2017.*
>
> *[2] Image super-resolution via iterative refinement. TPAMI 2022.*
>
> *[3] AirNet: All-in-One Image Restoration for Unknown Corruption. CVPR 2022.*
>
> *[4] PromptIR: Prompting for All-in-One Blind Image Restoration. NeurIPS 2023.*
>
> *[5] All in One Bad Weather Removal using Architectural Search. CVPR 2020.*

---

> > ### Author Response · Authors · 2023-11-23
> > **A Kind Reminder - Response period coming to an end**
> >
> > Dear Reviewer,
> >
> > This is a friendly reminder that the response period is coming to an end. We hope that our rebuttal and the revised manuscript have addressed most of your concerns, and we are happy to discuss any outstanding questions further.
> >
> > Thank you again for your comprehensive and positive review.
> >
> > Best regards,
> >
> > Authors

---

### Author Response · Authors · 2023-11-20
**General response**

We sincerely thank all reviewers for their detailed reviews and constructive comments. In the below response, we would like to summarise key clarifications in relation to some common review comments and concerns:

**Q1: Motivation for utilizing VLMs like CLIP.** The main motivation for leveraging VLMs in this work is to provide meaningful semantic instructions (e.g., degradation and clean content) to improve the restoration process. Although a common CNN can also classify degradation types (like in AirNet and PromptIR), its output contains far less information than that from a pretrained VLM. All experiments in the paper demonstrated that DA-CLIP consistently improves the image restoration performance of strong baseline models. Particularly, CLIP is pretrained on billions of web-scale text-image pairs which can provide more accurate visual representations and stronger generalization ability than vanilla-trained CNNs or vision Transformers. As illustrated in below Table 1, our method performs much better than other unified models on the unseen light rain dataset Rain100L, demonstrating the superiority of DA-CLIP. We provided more visual examples *in Figure 16 in the revised submission*.

**Table 1**. Comparison of different unified models on the unseen light rain Rain100L dataset.

|  Method  |  PSNR  |  SSIM  |  LPIPS  |  FID  |
|  ----  | ----  |  ----  | ----  |  ----  |
| AirNet | 30.07 | 0.935 | 0.114 | 45.56 |
| PromptIR | 32.77 | 0.950 | 0.081 | 32.31 |
| IR-SDE | 30.13 | 0.904 | 0.098 | 35.05 |
| Ours | **36.61** | **0.967** | **0.025** | **11.98** |

**Q2: DA-CLIP can be integrated also with MSE-based models to improve their performance.** We have shown the case of `NAFNet+DA-CLIP` in *Table 3 of the manuscript*, in which this variant outperforms the original NAFNet and even PromptIR by simply adding DA-CLIP embeddings. Here, we provide additional ablation experiments of applying DA-CLIP to NAFNet and PromptIR on unified restoration in Table 2 below for further clarification. For degradation-specific tasks, although there is no need to classify degradation types, we can still apply the HQ content embeddings of DA-CLIP to NAFNet to improve its performance, as in the experiments in Table 3 below.

**Table 2**. Ablation experiments of integrating DA-CLIP with NAFNet and PromptIR for *unified image restoration*. Note that PromptIR is built on Restormer with additional degradation prompt modules.

|  Method  |  PSNR  |  SSIM  |  LPIPS  |  FID  |
|  ----  | ----  |  ----  | ----  |  ----  |
| NAFNet | 26.34 | 0.847 | 0.159 | 55.68 |
| NAFNet + Degradation embedding | 27.02 | 0.856 | 0.146 | 48.27 |
| NAFNet + Degradation & Content embeddings | **27.22** | **0.861** | **0.145** | **47.94** |
| Restormer | 26.43 | 0.850 | 0.157 | 54.03 |
| PromptIR (Restormer + degradation prompt) | 27.14 | 0.859 | 0.147 | 48.26 |
| PromptIR + Content embedding | **27.26** | **0.861** | **0.145** | **47.75** |

**Table 3**. Experiments of integrating the DA-CLIP into NAFNet for *degradation-specific restoration*.

|  Deraining  |  PSNR  |  SSIM  |  LPIPS  |  FID  |  Low-light enhancement  |  PSNR  |  SSIM  |  LPIPS  |  FID  |
|  ----  | ----  |  ----  | ----  |  ----  |  ----  | ----  |  ----  | ----  |  ----  |
| NAFNet | 31.49 | 0.903 | 0.087 | 31.05 | NAFNet | 23.09 | 0.839 | 0.122 | 57.45 |
| NAFNet + DA-CLIP | **31.68** | **0.907** | **0.086** | **31.02** | NAFNet + DA-CLIP | **23.72** | **0.844** | **0.116** | **50.69** |

**Q3: DA-CLIP can benefit both unified and degradation-specific image restoration.** While our degradation embedding helps unified image restoration, the HQ content embedding of DA-CLIP can benefit the general restoration including all degradation-specific models. As we illustrated in *Table 2 and Figure 11 of the manuscript*, by simply integrating the predicted HQ content embedding into the network using cross-attention, our method significantly outperforms the baseline model (IR-SDE) across all degradation-specific tasks. In addition, the results in Table 3 above further demonstrated that DA-CLIP benefits degradation-specific restoration also for MSE-based approaches.

---

### Author Response · Authors · 2023-11-20
**Summary of Changes**

We appreciate all valuable feedback and comments from reviewers. Key updates in the revised submission include:

1. We have rewritten the introduction to clarify the concept of *universal image restoration*. We have also clarified the motivation for unified approaches.
2. In Section 2, we added a discussion on blind image restoration approaches (e.g., Real-ESRGAN, BSRGAN, StableSR, and DiffBIR), and we clarified that these methods are not universal since they are limited to a smaller set of degradation types (super-resolution, noise, and blur) and can't recover images from specific scenes such as rain, haze, shadow, etc.
3. For experiments:
	- We added the results of Restormer for unified image restoration (Table 3);
	- We provided two ablation experiments of applying DA-CLIP to MSE-based methods (NAFNet and PromptIR) on both unified (Table 3 and Table 6) and degradation-specific (Table 7) restoration tasks.
	- We tested our model on a light rain dataset (Table 5) and other real-world images (Figure 16) as out-of-distribution experiments.
4. For discussion and analysis:
	- We added an analysis of DA-CLIP with ground truth embeddings (Figure 9(b)).
	- We compared the HQ content embedding with the original CLIP image content embedding (Figure 9(c) and 9(d)).
	- We further added the comparison of directly applying text encoder generated embeddings (caption and degradation type) to the baseline (Figure 9(f) and 9(g)).
	- We provided an ablation experiment on the prompt degradation module (Figure 9(h)).
	- We added PromptIR and NAFNet in the model complexity comparison (Table 8) and reported the inference time for all approaches.
5. More experimental details (such as the prompt setting) were added in the corresponding sections and Appendix.

---

### Comment · Area_Chair_RXrB · 2023-11-20

Dear reviewers,

As the Author-Reviewer discussion period is going to end soon, please take a moment to review the response from the authors and discuss any further questions or concerns you may have.

Even if you have no concerns, it would be helpful if you could acknowledge that you have read the response and provide feedback on it.

Thanks,
AC

---

### Meta-Review · Area_Chair_RXrB · 2023-12-05

**Metareview:**

This paper proposes DA-CLIP for Multi-Task Image Restoration. In the proposed method, CLIP is used to provide representations of corruption as well as clean image to assist restoration. The proposed DA-CLIP achieves superior performance when compared to existing works. This paper receives a mixed rating of 3,6,6,6, and no consensus is reached during the discussion period. The AC has gone through the paper, reviews, and response, and thinks that the proposed work considers an interesting direction, bringing new insights to the community. Therefore an acceptance is recommended. The authors are advised to follow suggestions of reviewers, especially reviewer hnNQ, to tone down, avoiding the use of "universal" and "all-in-one". Also, please revise the paper according to the comment of the reviewers.

**Justification For Why Not Higher Score:**

While the proposed model demonstrates good performance, the concerns raised by reviewers about the generalizabilty and the necessity of using CLIP cannot be fully addressed. Therefore, the AC believes that a poster would be more suitable for the paper.

**Justification For Why Not Lower Score:**

The AC thinks that this paper considers an interesting direction and could provide insights for future works. Therefore, an acceptance is recommended.

---

### Decision · Program_Chairs · 2024-01-16

Accept (poster)